# Myocardial TRPC6-mediated $Zn^{2+}$ influx induces beneficial positive inotropy through β-adrenoceptors

Sayaka Oda[1,2,3,12], Kazuhiro Nishiyama [4,12], Yuka Furumoto[4],
Yohei Yamaguchi [5], Akiyuki Nishimura [1,2,3], Xiaokang Tang[1,2,3], Yuri Kato [4],
Takuro Numaga-Tomita[1,2,6], Toshiyuki Kaneko[5], Supachoke Mangmool [7],
Takuya Kuroda[8], Reishin Okubo[4], Makoto Sanbo [1], Masumi Hirabayashi [1],
Yoji Sato [8], Yasuaki Nakagawa[9], Koichiro Kuwahara [6], Ryu Nagata[10],
Gentaro Iribe [5], Yasuo Mori [11] & Motohiro Nishida[1,2,3,4]✉

Baroreflex control of cardiac contraction (positive inotropy) through sympathetic nerve activation is important for cardiocirculatory homeostasis. Transient receptor potential canonical subfamily (TRPC) channels are responsible for $α_1$-adrenoceptor ($α_1$AR)-stimulated cation entry and their upregulation is associated with pathological cardiac remodeling. Whether TRPC channels participate in physiological pump functions remains unclear. We demonstrate that TRPC6-specific $Zn^{2+}$ influx potentiates β-adrenoceptor (βAR)-stimulated positive inotropy in rodent cardiomyocytes. Deletion of *trpc6* impairs sympathetic nerve–activated positive inotropy but not chronotropy in mice. TRPC6-mediated $Zn^{2+}$ influx boosts $α_1$AR-stimulated $βAR/G_s$-dependent signaling in rat cardiomyocytes by inhibiting β-arrestin-mediated βAR internalization. Replacing two TRPC6-specific amino acids in the pore region with TRPC3 residues diminishes the $α_1$AR-stimulated $Zn^{2+}$ influx and positive inotropic response. Pharmacological enhancement of TRPC6-mediated $Zn^{2+}$ influx prevents chronic heart failure progression in mice. Our data demonstrate that TRPC6-mediated $Zn^{2+}$ influx with $α_1$AR stimulation enhances baroreflex-induced positive inotropy, which may be a new therapeutic strategy for chronic heart failure.

Baroreflex control of cardiac contractility, primarily left ventricular (LV) contractility (inotropy), through the autonomic nervous system is indispensable for maintaining constant blood pressure and cardiocirculatory homeostasis[1,2]. Blunting baroreflex-induced positive inotropy is a major cause of chronic heart failure, and boosting the positive inotropic response has received much attention as a potential therapeutic strategy[3,4]. Norepinephrine (NE) released from sympathetic nerve endings stimulates adrenoceptors (ARs) and augments positive inotropic and chronotropic effects. βARs are mainly expressed in the heart[5] and predominantly mediate augmentation of cardiac

contractility. Downregulation of βAR-dependent signaling is believed to be a major cause of heart failure, and several approaches to activate βAR signaling have been evaluated for treatment of heart failure, but none have succeeded. The binding affinity of βARs for NE is only 5% compared with that of isoproterenol (ISO), a βAR-selective artificial ligand[6], and a much higher concentration of NE compared with ISO is required to induce maximal cardiomyocyte contraction[7]. NE selectively binds to $α_1$AR in cardiomyocytes, but does not participate in the positive inotropic response. However, inhibition of cardiac $α_1$AR reportedly exacerbates heart failure in humans and mice[8,9], which

**Fig. 1 | Deletion of TRPC6 reduces positive inotropic effects in mouse hearts. a, b** Mice (129/Sv) were injected with hydralazine (Hyd; 0.5 mg/kg, 1.2 mL/h, *i.v.*). Time courses of increases in LV $\Delta$dP/dt max (**a**) and $\Delta$heart rate (**b**) induced by hydralazine administration (WT, $n = 9$; TRPC6$^{(-/-)}$, $n = 5$; TRPC3$^{(-/-)}$, $n = 5$); $P = 0.0019$ (6 min), 0.0092 (7 min), 0.0005 (8 min), 0.0231 (9 min), 0.0064 (10 min) in (**a**). Time courses of increases in LV $\Delta$dP/dt max (**c**), $\Delta$heart rate (**d**) and LV relaxation time $\tau$ (**e**) induced by $\beta$AR stimulation ($n = 6$ each group); $P = 0.0099$ (10 min), 0.0034 (12 min), 0.0010 (14 min), 0.0003 (16 min), 0.0003 (18 min), <0.0001 (20 min) in (**c**). $P = 0.0178$ (6 min), 0.0229 (8 min), 0.0120 (10 min), 0.0072 (12 min), 0.0241 (14 min), 0.0145 (20 min) in (**e**). Mice (129/Sv) were injected with isoproterenol (ISO, 10 pg/g/min, 0.05 mL/h, *i.v.*). Time courses of LV $\Delta$dP/dt max (**f**) and $\Delta$heart rate (**g**) following administration of hydralazine (0.5 mg/kg, 1.2 mL/h, *i.v.*) to mice (129/Sv) under a hyperglycemic condition [STZ (−), $n = 5$; STZ (+), $n = 7$]; $P = 0.0017$ (6 min), 0.0151 (7 min), 0.0006 (8 min), 0.0087 (9 min), 0.0007 (10 min) in (**f**). Data are shown as mean ± SEM. Significance was determined using two-way ANOVA followed by Sidak's comparison test. *$P < 0.05$; **$P < 0.01$.

implies that $\alpha_1$AR positively regulates baroreflex-induced $\beta$AR-dependent positive inotropy, but the molecular details are unknown.

Canonical subfamily members of transient receptor potential (TRPC) proteins are phospholipase C−linked receptor-operated cation channels in vertebrates[10]. Two diacylglycerol-sensitive TRPC isoforms, TRPC3 and TRPC6, are involved in pathological cardiac hypertrophy[11–14]. TRPC3 and TRPC6 channel activities have little effect on physiological Ca$^{2+}$ handling in normal cardiomyocytes, but increased expression levels of TRPC proteins are associated with activation of Ca$^{2+}$-dependent hypertrophic gene expression. Our previous studies using TRPC3-deficient mice have revealed that TRPC3 contributes to oxidative stress-related heart failure through a protein−protein interaction with NADPH oxidase (Nox) 2[15–17], but TRPC6-deficient mice revealed that TRPC6 negatively regulates TRPC3-Nox2 interaction and oxidative stress[15,18]. This indicates that although TRPC6 and TRPC3 have share amino acid sequence homology and a common activation mechanism, the pathophysiological role of TRPC6 in the heart is different from that of TRPC3.

Compared with TRPC3, the TRPC6 channel has a unique characteristic, namely heavy metal ion permeability[19,20]. Deletion of TRPC6 disrupts Zn$^{2+}$ homeostasis and pregnancy outcomes in mice[19] and stable expression of TRPC6 in HEK293 cells allows Zn$^{2+}$ entry, leading to a larger intracellular Zn$^{2+}$ pool[20]. Almost all intracellular Zn$^{2+}$ binds to proteins or is distributed in organelles and vesicles. Elevation of labile Zn$^{2+}$ associated with an increase in the intracellular Zn$^{2+}$ concentration is attracting attention as a second messenger[21–23]. Zn$^{2+}$ has a role in maintaining cardiac redox and energy homeostasis by competing with Fe and Cu for uptake, both of which cause oxidative stress[24], which suggests the importance of nutritional Zn$^{2+}$ supplementation in

patients with heart failure[25]. In addition, the agonist-binding affinity of $\beta_2$AR is enhanced by intracellular Zn$^{2+}$ through allosteric modulation of $\beta_2$AR[26,27]. Therefore, in this work we assessed whether TRPC6-mediated Zn$^{2+}$ influx physiologically contributes to the sympathetic nerve−activated cardiac positive inotropic response by enhancing $\beta$AR/G$_s$-dependent signaling and whether it has a beneficial effect on chronic heart failure.

## Results
### TRPC6 participates in baroreflex positive inotropy in mouse hearts

Because TRPC forms a receptor-operated cation channel, we first examined whether TRPC6 participates in autonomic nervous regulation of cardiac functions in mice. Baroreflex-dependent cardiac responses can be experimentally induced by hydralazine, a vasodilator, through NE release from sympathetic nerve endings[28]. Hydralazine administration induced a compensatory positive inotropic effect represented by changes in LV dP/dt$_{max}$ in wildtype (WT) and TRPC3-deficient [TRPC3$^{(-/-)}$] mice (Fig. 1a). This positive inotropic response was impaired in TRPC6-deficient [TRPC6$^{(-/-)}$] mice. Conversely, the hydralazine-induced compensatory increase in heart rate (chronotropic effect) was not different among WT, TRPC3$^{(-/-)}$ and TRPC6$^{(-/-)}$ mice (Fig. 1b). Because hydralazine decreased both systolic and diastolic blood pressures in both WT and TRPC6$^{(-/-)}$ mice (Supplementary Fig. 1a), TRPC6 deficiency may attenuate baroreflex control of cardiac positive inotropy downstream of NE release.

We next examined whether TRPC6 deficiency attenuates $\beta$AR-stimulated positive inotropy. ISO-induced positive inotropy, chronotropy and lusitropy were similarly induced in WT and TRPC3$^{(-/-)}$ mice,

while ISO-induced positive inotropy and lusitropy, but not chronotropy, were impaired in TRPC6$^{(-/-)}$ mice (Fig. 1c–e). Conversely, activation of the parasympathetic nervous system mimicked by acetylcholine administration exerted negative inotropic and chronotropic effects to similar extents in all three groups (Supplementary Fig. 1b). Activation of the sympathetic nervous system mimicked by NE administration caused transient increases in blood pressure to the same extent in WT, TRPC3$^{(-/-)}$ and TRPC6$^{(-/-)}$ mice (Supplementary Fig. 1c). Basal catheter parameters of WT, TRPC6$^{(-/-)}$ and TRPC3$^{(-/-)}$ mice were similar (Supplementary Table 1). The protein and mRNA expression levels of βAR were similar between WT and TRPC6$^{(-/-)}$ hearts (Supplementary Fig. 1d, e).

We next examined whether TRPC6 upregulation enhances cardiac positive inotropy. We have previously reported that streptozotocin (STZ)-induced hyperglycemia increases TRPC6 expression in the heart[18]. In STZ-injected hyperglycemic mice, the baroreflex positive inotropy induced by hydralazine was markedly enhanced while positive chronotropy was not enhanced compared with that in normal mice (Fig. 1f, g, Supplementary Table 2, and Supplementary Fig. 1f). These results suggest that TRPC6 selectively potentiates βAR-dependent positive inotropy induced by sympathetic nervous system activation.

### TRPC6 colocalizes with β₁AR in LV cardiomyocytes

Because TRPC6 is widely distributed in various cell types, which include cardiomyocytes and sinoatrial node (SAN) cells, we hypothesized that a short physical distance between TRPC6 and βAR is crucial for the selective involvement of TRPC6 in βAR-induced cardiac positive inotropy[29]. Colocalization of TRPC6 protein with β₁AR protein in close proximity (<40 nm distances) was visualized as puncta using a proximity ligation assay (PLA). The number of PLA puncta observed on T-tubules and sarcolemma was significantly higher in LV cardiomyocytes than in hyperpolarization-activated cyclic nucleotide-gated channel 4-positive SAN cells (Fig. 2a, b and Supplementary Fig. 2a, b). Additionally, PLA puncta were abundantly detected in T-tubules of isolated cardiomyocytes (Fig. 2c). The PLA puncta of TRPC6 and β₁AR were completely diminished in TRPC6$^{(-/-)}$ cardiomyocytes. Furthermore, β₁AR signals disappeared upon β₁AR knockdown (Supplementary Fig. 2c–e), supporting the specificity of the anti-β₁AR antibody. Thus, colocalization of TRPC6 and β₁AR in LV cardiomyocytes (abundant T-tubules), but not SAN cells (rare T-tubules), might be a reason why TRPC6 selectively participates in baroreflex positive inotropy and lusitropy, but not chronotropy, in mouse hearts. In addition, TRPC6 and β₁AR co-localized completely on the plasma membrane, and FLAG-tagged TRPC6 were co-immunoprecipitated with GFP-fused β₁AR (Supplementary Fig. 2f). Thus, TRPC6 physically forms protein complex with β₁AR.

The number of PLA puncta was significantly increased at the base of hyperglycemic hearts compared with normal hearts (Fig. 2d, e), but was not increased at the apex region. The reason for the different PLA puncta distribution between the apex and base is unknown, but this difference might explain the different responsibility of NE in Takotsubo cardiomyopathy, triggered by increased sympathetic nerve activity and NE spillover, which is characterized by transient apical ballooning in the LV apex region and hypercontraction in the base region[30]. These results suggest that hyperglycemia-induced TRPC6 upregulation at the base of the heart contributes to augmentation of baroreflex cardiac positive inotropy.

### TRPC6 positively regulates βAR/Gₛ-mediated cardiomyocyte contraction

We next examined whether TRPC6 deletion impairs βAR-stimulated enhancement of contractility in isolated adult mouse cardiomyocytes. The increase in sarcomere length shortening amplitude and the decrease in the relaxation decay by ISO stimulation were significantly suppressed in TRPC6$^{(-/-)}$ cardiomyocytes (Fig. 3a, b). ISO-stimulated enhancement of Ca$^{2+}$ transient evoked by electric field stimulation was also impaired in TRPC6$^{(-/-)}$ cardiomyocytes (Fig. 3c, d).

βAR preferentially couples with heterotrimeric Gₛ protein and enhances Ca$^{2+}$ handling through cyclic adenosine monophosphate (cAMP) production. Changes in the intracellular cAMP concentration in neonatal rat cardiomyocytes (NRCMs) were visualized using a Förster resonance energy transfer biosensor[31]. βAR-stimulated cAMP production was suppressed in TRPC6-silenced NRCMs (Fig. 3e, f). The mRNA expression levels of adenylate cyclases and G proteins were not changed between WT and TRPC6-silenced NRCMs or among WT, TRPC6$^{(-/-)}$ and TRPC3$^{(-/-)}$ mouse hearts (Supplementary Fig. 3a, b). These results suggest that TRPC6 directly potentiates βAR-stimulated cAMP production and Ca$^{2+}$ handling in LV cardiomyocytes.

### The α₁AR-TRPC6-Zn$^{2+}$ axis enhances βAR-Gₛ signaling

Compared with the TRPC3 channel, the TRPC6 channel has a unique characteristic of Zn$^{2+}$ permeability[32,33]. We next investigated whether TRPC6-mediated Zn$^{2+}$ influx participates in βAR-stimulated cardiac positive inotropy. Treatment with 2,2′-dithiodipyridine (DTDP) oxidizes Zn$^{2+}$-binding proteins and induces Zn$^{2+}$ release from the intracellular Zn$^{2+}$ pool[34]. The pooled Zn$^{2+}$ amount in DTDP-treated TRPC6$^{(-/-)}$ cells was smaller than those in DTDP-treated WT and TRPC3$^{(-/-)}$ cells (Fig. 4a–e). This supports the previous reports showing that TRPC6 expression per se is important to maintain basal intracellular Zn$^{2+}$ homeostasis[19,20]. The mRNA expression levels of Zn$^{2+}$ transporters, ZIPs and ZnTs, metallothioneins and endoplasmic reticulum stress markers were similar among WT, TRPC3$^{(-/-)}$ and TRPC6$^{(-/-)}$ mouse hearts (Supplementary Figs. 3d, e and 4a).

Because NE released from sympathetic nerve endings preferentially stimulates αAR in cardiomyocytes[35] and Gq-coupled αAR stimulation increases TRPC6 channel activity[36,37], we hypothesized that αAR-stimulated TRPC6 enhances the βAR-dependent positive inotropic response through Zn$^{2+}$ influx. TRPC6 interacts with α₁AAR in cardiomyocytes[38]. Stimulation of α₁AAR with NE induced a marked increase of the intracellular Zn$^{2+}$ concentration in TRPC6- and α₁AAR-expressing HEK293 cells compared with that in cells expressing TRPC3 and α₁AAR (Fig. 4f and Supplementary Fig. 4b). The NE-induced TRPC6-dependent increase in the intracellular Ca$^{2+}$ concentration was observed in α₁AAR-expressing cells, but not in β₁AR-expressing cells (Supplementary Fig. 4c, d), which indicated that α₁AAR predominantly mediates NE-induced TRPC6 channel activation.

We compared the amino acid sequences of Zn$^{2+}$-permeable TRPC6 and Zn$^{2+}$-impermeable TRPC3 and found TRPC6-specific sequences (N$^{615}$YN$^{617}$) close to the pore region (Supplementary Fig. 4e). Mutation of NYN to KYD, the same sequence as TRPC3 and TRPC7, abolished the NE-induced TRPC6-dependent Zn$^{2+}$ influx (Fig. 4g and Supplementary Fig. 4f). Conversely, the muscarinic receptor-stimulated Ca$^{2+}$ influx in TRPC6(KYD)-expressing cells was equivalent to that in TRPC6(WT)-expressing cells (Supplementary Fig. 4g–i), which suggested that the NYN sequence is critical for Zn$^{2+}$ permeability of TRPC6.

In NRCMs, the NE-induced increase in cAMP production was partially but significantly suppressed by prazosin (α₁AR inhibitor) and N,N,N′,N′-tetrakis(2-pyridylmethyl)ethylenediamine (TPEN, Zn$^{2+}$ chelator) and completely suppressed by propranolol (β$_{1/2}$AR inhibitor) (Fig. 4h). Conversely, treatment with an αAR agonist (methoxamine) for 24 h significantly enhanced ISO-induced cAMP production (Fig. 4i), which was canceled by TRPC6 knockdown (Fig. 4j). TRPC6 channel activators, 2-[4-(2,3-dimethylphenyl)-piperazin-1-yl]-N-(2-ethoxyphenyl)acetamide (PPZ2)[39] and GSK1702934A, significantly increased the amount of intracellular Zn$^{2+}$ through TRPC6 channels (Supplementary Fig. 4j–m). Treatment of adult cardiomyocytes with PPZ2 increased the ISO-stimulated cAMP production (Fig. 4k). Conversely, treatment with TPEN decreased the ISO-stimulated cAMP

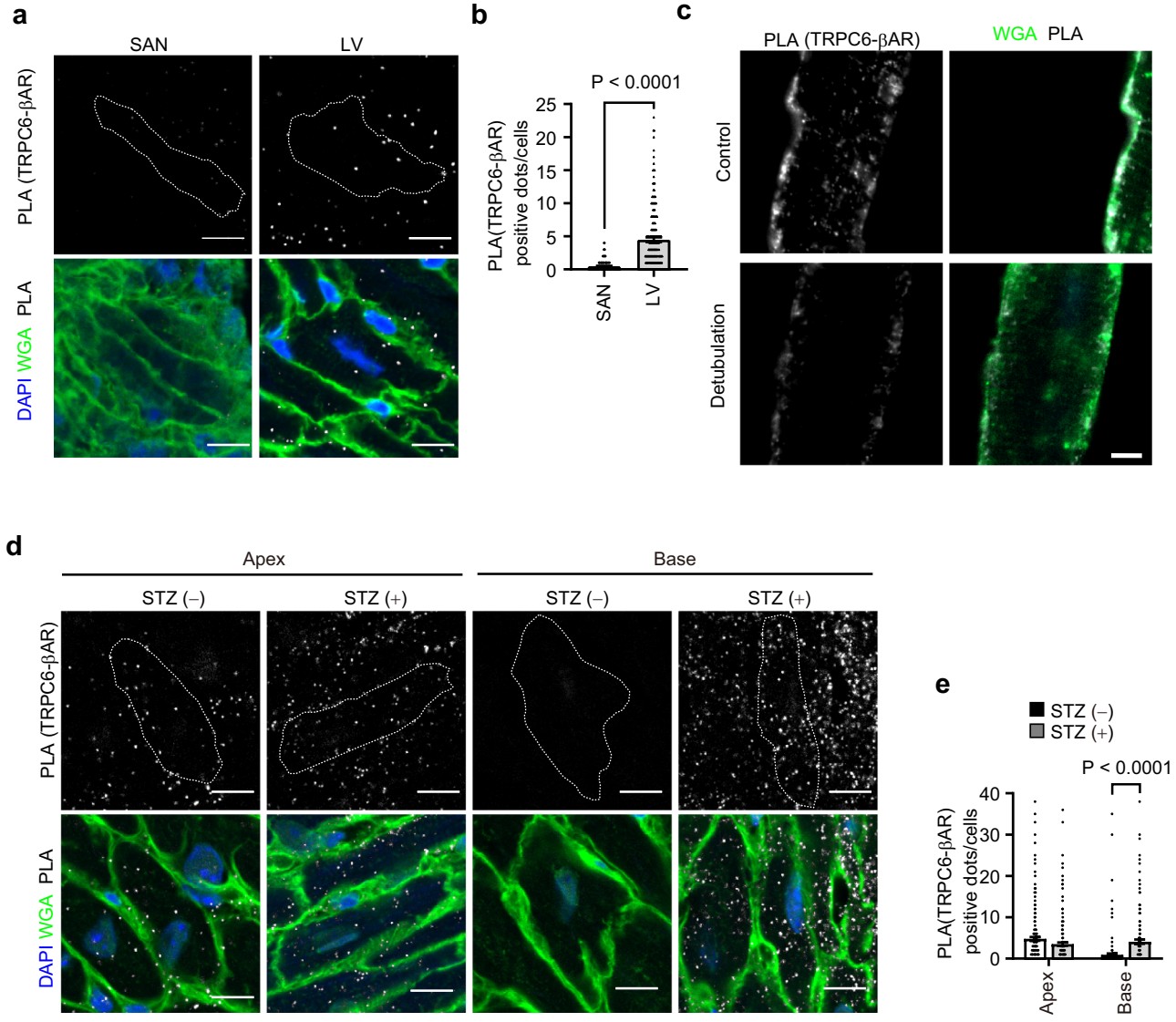

**Fig. 2 | TRPC6 exists in close proximity to β₁AR in the left ventricle (LV) myocardium. a** Representative images of the Duolink proximity ligation assay (PLA) between TRPC6 and β₁AR. Cells from sinoatrial node (SAN) (left) and LV (right). PLA signals are shown as white spots counterstained with wheat germ agglutinin (WGA: green) and DAPI (blue). In the LV group, images were captured from the apex area. Scale bar, 10 μm. **b** Number of PLA signals for each cell from SAN or LV. SAN, $n = 180$ cells; LV, $n = 180$ cells. **c** Representative PLA images of TRPC6 and β₁AR in adult cardiomyocytes with or without formamide (1.5 M). PLA signals are colored by white spots in the merged image and counterstained with WGA (green in merge images). Scale bar, 5 μm. **d** Representative images of PLA of TRPC6 and β₁AR. Cells from the apex area (left) and base area (right) in mice with or without STZ treatment. PLA signals are shown as white spots counterstained with WGA (green) and DAPI (blue). Scale bar, 10 μm. **e** Number of PLA signals for each cell from the apex and base area in mice with or without STZ treatment. Apex STZ(−), $n = 239$ cells; Apex STZ(+), $n = 236$ cells; Base STZ(−), $n = 179$ cells; Base STZ(+), $n = 169$ cells. Data are shown as mean ± SEM. **$P < 0.01$ by the unpaired t-test (**b**, two-sided) or two-way ANOVA followed by Sidak's post-hoc test (**e**). Images are representative of three independent experiments (**a**, **c**, **d**).

production (Fig. 4l). The increase in cAMP production by ISO stimulation was significantly reduced in TRPC6^(KYD/KYD) cardiomyocytes compared with WT cells (Fig. 4m and Supplementary Fig. 5). Treatment of HEK293 cells with $Zn^{2+}$ enhanced the NE-induced interaction between β₁AR and Gα$_s$ (Supplementary Fig. 6a). These results suggest that αAR-mediated TRPC6 activation potentiates NE-induced cAMP production in cardiomyocytes in a $Zn^{2+}$-dependent manner.

**TRPC6-mediated $Zn^{2+}$ influx negatively regulates β-arrestin-dependent βAR internalization**

After βAR stimulation by NE release, βAR is phosphorylated by G protein-coupled receptor kinases, which leads to βAR internalization through a β-arrestin (βArr)-dependent pathway[40]. ISO-stimulated internalization of GFP-fused β₁AR was significantly suppressed in TRPC6-expressing HEK293 cells, but not in control cells or cells

expressing TRPC3 or the pore-dead dominant negative mutant of TRPC6 (TRPC6-DN; in which the LFW motif conserved in the putative pore region of all TRPCs was substituted with AAA) (Fig. 5a, b). ISO-induced plasma membrane translocation of YFP-tagged βArr2 was also suppressed in TRPC6-expressed cells, but not in TRPC6-DN- or TRPC3-expressing cells (Fig. 5c−e). The number of PLA puncta for the interaction between β₁AR and βArr2 was significantly reduced in TRPC6^(−/−) cardiomyocytes compared with that in WT cells (Fig. 5f, g). These results indicate that TRPC6 negatively regulates βArr-dependent βAR internalization upon ISO stimulation. Furthermore, TRPC6-mediated inhibition of βArr recruitment to βAR was attenuated by a $Zn^{2+}$ chelator (TPEN) but not a $Ca^{2+}$ chelator (EGTA) (Supplementary Fig. 6b). TRPC6 does not regulate βArr-dependent angiotensin II type 1 receptor internalization upon angiotensin II stimulation (Supplementary Fig. 6c), suggesting a specific functional coupling of TRPC6 with βARs.

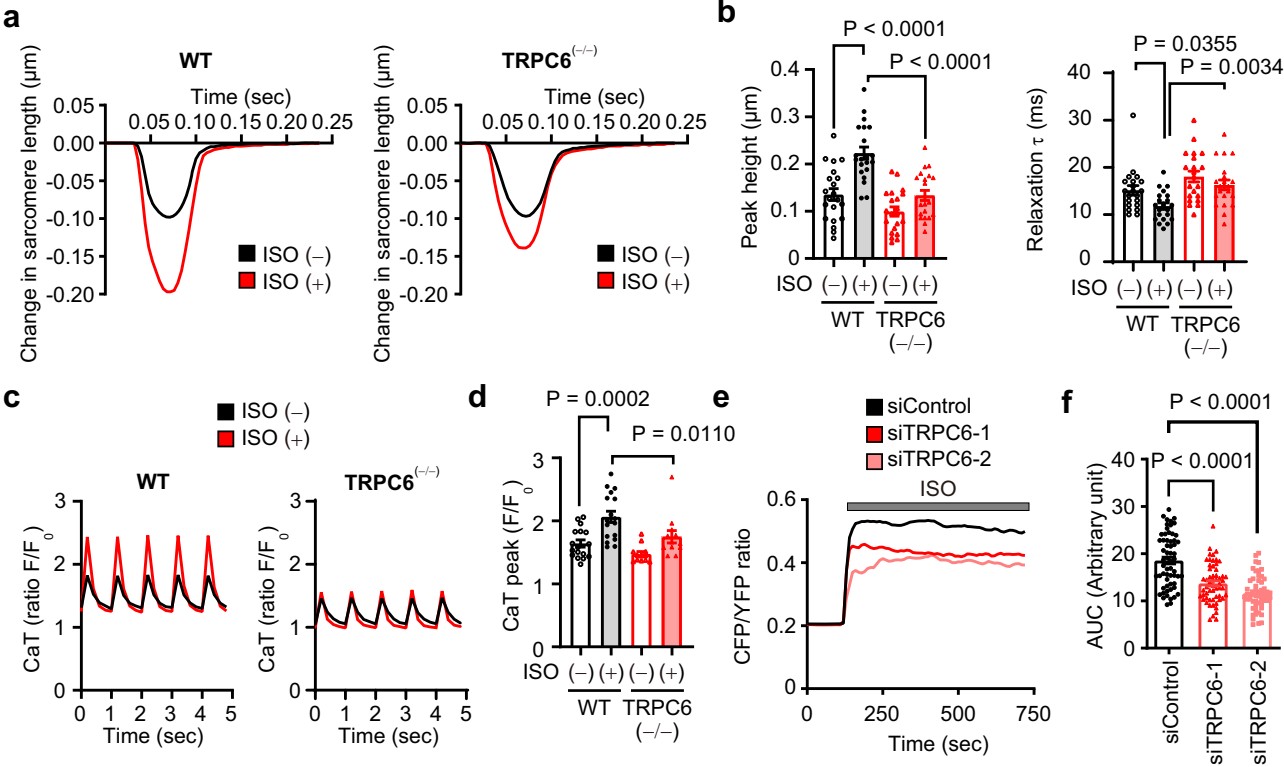

**Fig. 3 | TRPC6 deletion reduces βAR-stimulated contraction in adult mouse cardiomyocytes. a, b** Representative traces of sarcomere length changes (**a**) and summarized results of peak heights and relaxation time τ (**b**) induced by ISO stimulation (10 nM). Isolated mouse (129/Sv) cardiomyocytes were excited by field stimulation at 4 Hz. WT, $n = 21$ cells; TRPC6$^{(-/-)}$, $n = 21$ cells. Representative traces of calcium transients (CaT) (**c**) and summarized results of peak amplitudes of CaT (**d**). Fluo4-loaded cardiomyocytes were stimulated with ISO (10 nM) under field stimulation of 10 msec, 50 V at 1 Hz. WT, $n = 17$ cells; TRPC6$^{(-/-)}$, $n = 12$ cells. **e, f**
Average cAMP production in control and TRPC6-silenced NRCMs (**e**). NRCMs that expressed the Epac-based FRET biosensor were stimulated with ISO (1 μM). Increases in intracellular cAMP concentrations are shown as the area under the curve (AUC) (**f**). siControl, $n = 60$ cells; siTRPC6-1, $n = 57$ cells; siTRPC6-2, $n = 55$ cells. Data are shown as mean ± SEM. *$P < 0.05$; **$P < 0.01$ using one-way ANOVA followed by Tukey's post-hoc test (**f**) or two-way ANOVA followed by Sidak's post-hoc test (**b, d**).

ISO-induced βArr2 translocation was suppressed in TRPC6 (WT)-expressing cells but not in TRPC6 (KYD)-expressing cells (Fig. 5h). The mRNA expressions of βArr2 in WT, TRPC3$^{(-/-)}$ and TRPC6$^{(-/-)}$ mice and protein expressions of GRK5/6 in WT, TRPC3$^{(-/-)}$, TRPC6$^{(-/-)}$ and TRPC3/6$^{(-/-)}$ mice were not significantly different (Supplementary Fig. 3b, c). These results strongly suggest that TRPC6-mediated Zn$^{2+}$ influx negatively regulates βArr-dependent βAR internalization.

### Activation of TRPC6-mediated Zn$^{2+}$ influx improves chronic heart failure

We finally examined whether activation of the α$_1$AR-TRPC6-βAR axis improves positive inotropy and chronic heart failure through TRPC6-mediated Zn$^{2+}$ influx. Treatment of NRCMs with TPEN decreased the NE-stimulated cardiomyocyte contraction force (Supplementary Fig. 7a). α$_1$AR interacts with TRPC6 via a scaffolding protein, Snapin[37], and βAR and TRPC6 proteins preferentially interact with postsynaptic density protein PSD-95 through its PDZ-binding motif[41]. We constructed a plasmid expressing a Snapin-PDZ-linker to induce α$_1$AR-TRPC6-βAR triple complex formation. We found that expression of a Snapin-PDZ-linker significantly enhanced the NE-stimulated cardiomyocyte contraction in NRCMs (Supplementary Fig. 7b). Treatment of NRCMs with PPZ2 increased the NE-stimulated cardiomyocyte contraction force in a Zn$^{2+}$-dependent manner (Supplementary Fig. 7c). Long-time treatment with PPZ2 had no effect on TRPC6 localization (Supplementary Fig. 7d). suggesting no inverse agonist activity for TRPC6. Overexpression of the Snapin-PDZ-linker peptide significantly increased the number of PLA puncta of TRPC6 and β$_1$AR in NRCMs

(Supplementary Fig. 7e). Treatment with PPZ2 enhanced cardiomyocyte contraction under electrical stimulation where βARs are partially stimulated[42] and hydralazine-induced positive inotropy in a TRPC6 inhibition–dependent manner (Supplementary Fig. 7 f–h, Fig. 6a).

We investigated the in vivo cardioprotective effect of PPZ2 against chronic heart failure using three different mouse models: the transverse aortic constriction (TAC), myocardial infraction (MI) and muscle LIM protein–deficient (MLP-KO) dilated cardiomyopathic mice. Treatment of TAC-operated mice with PPZ2 significantly improved LV dysfunctions and structural remodeling, such as myocardial hypertrophy and interstitial fibrosis, in a TRPC6-dependent manner (Fig. 6b–d, Supplementary Fig. 8a and Supplementary Tables 3 and 4). TRPC6 is upregulated in mouse hearts with pathological hypertrophy[14], and protein kinase G (PKG) is an anti-hypertrophic factor that is activated by βAR stimulation through a nitric oxide synthase 3-dependent pathway[43,44]. PPZ2 treatment significantly increased intracellular Zn$^{2+}$ amounts and PKG activity in TAC-operated WT hearts, and these increases were not observed in TAC-operated TRPC6$^{(-/-)}$ hearts (Fig. 6e–g). PPZ2 treatment also improved LV dysfunction, cardiomyocyte hypertrophy and fibrosis in mice after MI and in MLP-KO mice (Fig. 6h, i, Supplementary Fig. 8b–g and Supplementary Tables 5–8). These results strongly suggest that pharmacological activation of TRPC6 improves chronic heart failure in mice.

## Discussion

Various lines of evidence have suggested that TRPC3 and TRPC6 channels are therapeutic targets of pathological cardiac

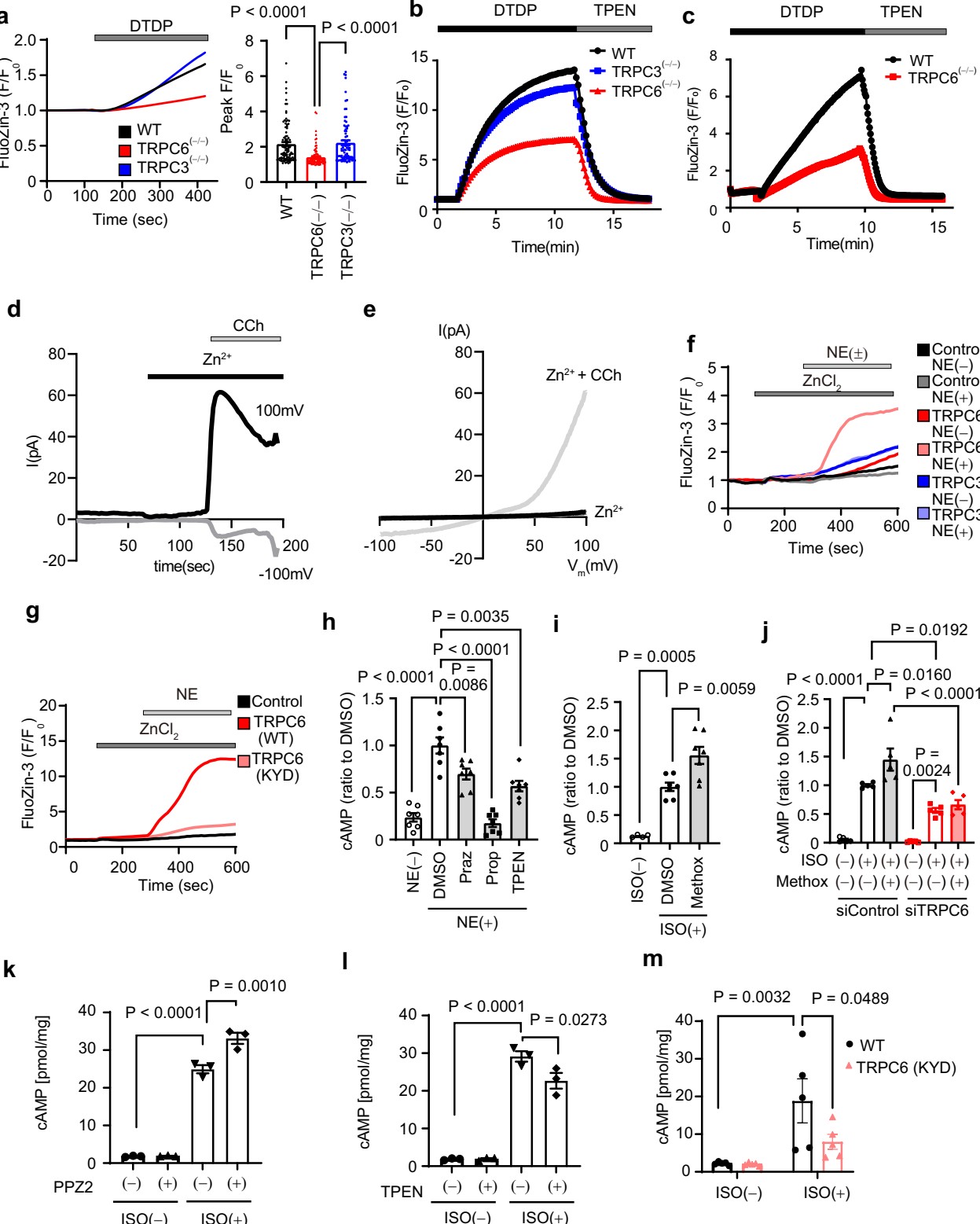

remodeling[15,16,18,45]. Several small compounds that inhibit TRPC3/6 channel activities directly or indirectly prevent pressure overload-induced heart failure in mice[13,46,47]. Conversely, TRPC6 expressed in cardiac fibroblasts is physiologically necessary for ischemic scar formation after MI[48]. Because TRPC6-deficient mice show no apparent cardiac phenotype, the physiological role of TRPC6 in cardiomyocytes has been unclear. Here, we revealed the physiological significance of TRPC6 in myocardial $Zn^{2+}$ homeostasis required

for beneficial cardiac positive inotropy (Fig. 7). How $Zn^{2+}$ enhances βAR-$G_s$ signaling can be explained in part by a previous report showing that $Zn^{2+}$ interacts with βAR directly and enhances ligand affinity and accumulation of intracellular cAMP[26]. $Zn^{2+}$ binds to the 269th histidine among all the histidines in human β2AR, and 225th glutamate and 265th cysteine in the cytoplasmic region supports the allosteric $Zn^{2+}$ binding to β2AR, which increases ligand affinity[27], but these three amino acids are not conserved in human β1AR. Another

**Fig. 4 | The $\alpha_1$AR-TRPC6-Zn$^{2+}$ axis enhances $\beta$AR-G$_s$ signaling. a** Time courses and peak intensities of FluoZin-3 fluorescence in adult mouse (129/Sv) cardiomyocytes. Cardiomyocytes were treated with DTDP (100 $\mu$M). WT, $n = 98$ cells; TRPC6$^{(-/-)}$, $n = 86$ cells; TRPC3$^{(-/-)}$, $n = 84$ cells. **b** Increase of intracellular FluoZin-3 fluorescence evoked by DTDP in mouse (129/Sv) splenocytes. TPEN chelates intracellular free Zn$^{2+}$. WT, $n = 300$ cells; TRPC6$^{(-/-)}$, $n = 240$ cells; TRPC3$^{(-/-)}$, $n = 170$ cells. **c** Increase of FluoZin-3 fluorescence evoked by DTDP in mouse (129/Sv) smooth muscle cells. WT, $n = 67$ cells; TRPC6$^{(-/-)}$, n = 67 cells. Representative time courses (**d**) and leak-subtracted I–V relationships (**e**) of carbachol (CCh, 100 $\mu$M)-induced TRPC6 currents in TRPC6-EGFP-expressing HEK293 cells with ZnCl$_2$ (10 mM). **f** Time courses of changes in intracellular Zn$^{2+}$ concentrations in response to NE stimulation in TRPC- and $\alpha_1$AR-expressing HEK293 cells. **g** Time courses of changes in intracellular Zn$^{2+}$ concentrations in response to NE stimulation in TRPC6 (WT or KYD)- and $\alpha_1$AR-expressing HEK293 cells. **h** Changes in intracellular cAMP concentrations after treatment with prazosin (Praz; 10 $\mu$M), propranolol (Prop;

10 $\mu$M) or TPEN (25 $\mu$M) in NRCMs. NRCMs were treated with prazosin or propranolol for 25 h or with TPEN for 1 h before NE stimulation (1 $\mu$M, 30 min). ZnCl$_2$ (50 $\mu$M) was added to the culture medium. $n = 7$ each group. **i** Intracellular cAMP changes in response to $\alpha_1$AR stimulation in NRCMs. NRCMs were treated with methoxamine (Methox, 1 $\mu$M) for 24 h before ISO stimulation (1 $\mu$M, 30 min). ZnCl$_2$ (50 $\mu$M) was added in the culture medium. $n = 7$ each group. **j** Requirement of TRPC6 for $\alpha_1$AR-induced enhancement of $\beta$AR-stimulated cAMP production in NRCMs. $n = 5$ each group. **k** Effect of PPZ2 (30 $\mu$M) on ISO (10 nM)-stimulated cAMP production in adult mouse (C57BL/6J) cardiomyocytes ($n = 3$ each group). **l** Effect of TPEN on ISO-stimulated cAMP production in adult mouse (C57BL/6J) cardiomyocytes ($n = 3$ each group). **m** ISO-stimulated cAMP production in WT and TRPC6$^{(KYD/KYD)}$ adult mouse (C57BL/6J) cardiomyocytes ($n = 5$ each group). Data are shown as mean ± SEM. *$P < 0.05$; **$P < 0.01$ using one-way ANOVA followed by Tukey's post-hoc test (**a**, **h**–**l**) or two-way ANOVA followed by Sidak's post-hoc test (**m**).

Zn$^{2+}$-binding candidate is G protein-coupled receptor kinase (GRK)-interacting protein 1, which negatively regulates GRK/$\beta$-arrestin-mediated $\beta$AR internalization[49]. Thus, several Zn$^{2+}$-binding proteins may cooperatively contribute to potentiation of $\beta$AR/G$_s$-dependent signaling.

Heterologous regulation of G protein-coupled receptor signaling is mediated through post-translational modifications[50] and direct protein–protein interactions[51,52]. We revealed a positive link between $\beta$AR and $\alpha_1$AR through TRPC6-mediated Zn$^{2+}$ influx. TRPC channels are thought to play a fundamental role in the amplification of G$_q$PCR signaling pathways[53]. We suggest that TRPC6 enhances G$_s$ protein-coupled $\beta$AR signaling in rodent hearts through Zn$^{2+}$ influx. This concept will provide a new therapeutic strategy for chronic heart failure by focusing on potentiation of baroreflex-induced $\beta$AR-dependent positive inotropy. However, $\beta$-blockers are given to most heart failure patients, and we demonstrated that propranolol antagonizes the $\alpha_1$AR-TRPC6-Zn$^{2+}$ axis (Fig. 4). This implies that the beneficial positive inotropic effect by TRPC6 activator is not expected in heart failure patients treated with $\beta$-blockers, but TRPC6-mediated Zn$^{2+}$ influx may help reduce the load on the heart. A recent report showing that Zn$^{2+}$ drives vasorelaxation by acting in sensory nerves, endothelium and smooth muscle[54] also supports the physiological significance of intracellular Zn$^{2+}$ as a new therapeutic target. We have previously reported that TRPC6-mediated local Ca$^{2+}$ signaling protects the heart against oxidative stress in cardiac cells[55]. Our results do not completely rule out the effect of TRPC6-mediated Ca$^{2+}$ influx, but strongly suggest that $\alpha_1$AR-mediated Zn$^{2+}$ mobilization through TRPC6 channels potentiates NE-induced cardiac positive inotropy, which may lead to improvement of chronic heart failure.

In addition to TRPC6, other TRPs permeate Zn$^{2+}$, such as TRPM3/6/7, TRPA1, TRPV6 and TRPML1[33]. These Zn$^{2+}$-permeable TRP channels possess metal ion–sensitive charged amino acids in their pore region, whereas diacylglycerol-responsive TRPC3/6/7 channels do not have these conserved amino acids. We found that TRPC6 lacks charged amino acids (K and D) near the pore region that contains the LFW motif, which are conserved in Zn$^{2+}$ impermeable TRPC3 and TRPC7 (Supplementary Fig. 4e). Mutation of these amino acids (NYN to KYD) caused loss of Zn$^{2+}$ permeability of the TRPC6 channel. We therefore speculate that removing the charged amino acids close to the pore region may be important for TRPC6 to permeate metal ions that include Zn$^{2+}$. Because the expression levels of these metal ion–permeable TRP channels are lower than that of TRPC6 and the TRPC subfamily proteins are the putative entities of receptor-operated cation channels, they likely have no contribution to NE-induced Zn$^{2+}$ influx. Notably, we demonstrated that increased TRPC6 channel activity or enhanced $\alpha$AR-TRPC6-$\beta$AR coupling improves $\beta$AR-dependent cardiac positive inotropy and chronic heart failure in mice, which suggests a novel pathophysiological role of TRPC6 in the heart. A more interesting finding is that the activation of TRPC6-

mediated Zn$^{2+}$ influx increases anti-hypertrophic PKG signaling specifically in the pathologically hypertrophied hearts (Fig. 6). Although the molecular details are unclear, we hypothesize that the PPZ2-induced PKG activation is a compensatory negative feedback mechanism to attenuate excess stress-stimulated contractility and arrhythmia through TRPC6 in dystrophic hearts[44].

In conclusion, we first revealed the physiological role of the TRPC6 channel in $\beta$AR-dependent positive inotropy in mouse hearts. Although TRPC6 has been thought to be a mediator of pathological cardiac remodeling, we demonstrated that the $\alpha_1$AR-stimulated Zn$^{2+}$ influx through the TRPC6 channel induced beneficial positive inotropy. The enhancement of TRPC6-specific Zn$^{2+}$ influx will be a new therapeutic target of chronic heart failure.

## Methods
### Reagents and antibodies
We obtained (−)-isoproterenol hydrochloride (ISO, I6504), (±)-norepinephrine (+)-bitartrate salt (NE, A0937), 2,2′-dithiodipyridine (DTDP, D5767), prazosin hydrochloride (P7791), (±)-propranolol hydrochloride (P0884), methoxamine hydrochloride (M6524) and streptozotocin (STZ, S0130) from Sigma Aldrich (Darmstadt, Germany). Fluo4-AM (F311) and N,N,N′,N′-tetrakis(2-pyridylmethyl)ethylenediamine (TPEN, 343-05401) were purchased from Dojindo (Kumamoto, Japan). Fluo-Zin™–3 AM (F24195) was from Thermo Fisher (Hampton, New Hampshire, USA). We purchased 1-hydrazinophthalazine hydrochloride (hydralazine hydrochloride, H0409) from Tokyo Chemical Industry (Tokyo, Japan). The antibodies used in this study are listed in Supplementary Table 9.

### Animals
All protocols using mice and rats were reviewed and approved by the ethics committees at the National Institutes of Natural Sciences and Kyushu University and carried out in accordance with the committee guidelines (approval no. A22-171, A20-040, A21-072, 22A026). We obtained 129/Sv mice with homozygous deletion of genes encoding TRPC6 and TRPC3 from the Comparative Medicine Branch, National Institute of Environmental Health Sciences (Research Triangle Park, NC, USA). Genotyping was performed as previously described[56,57]. Sprague–Dawley rats were purchased from Japan SLC, Inc (Shizuoka, Japan). TRPC6$^{(KYD/KYD)}$ C57BL/6J mice were generated using the CRISPR/cas9 system (Supplementary Figure 5). Genotyping of TRPC6$^{(KYD/KYD)}$ mice was conducted by PCR and DNA product sequencing. All mice and rats were housed in individually ventilated cages, with aspen wood chip bedding, in groups of three animals per cage and kept under controlled environmental conditions (specific-pathogen-free area, 12-hr light/12-hr dark cycle, room temperature 21–23 °C, and humidity 50–60%) with free access to standard laboratory food pellets (CLEA Rodent Diet CE-2, CLEA Japan, Tokyo, Japan) and water.

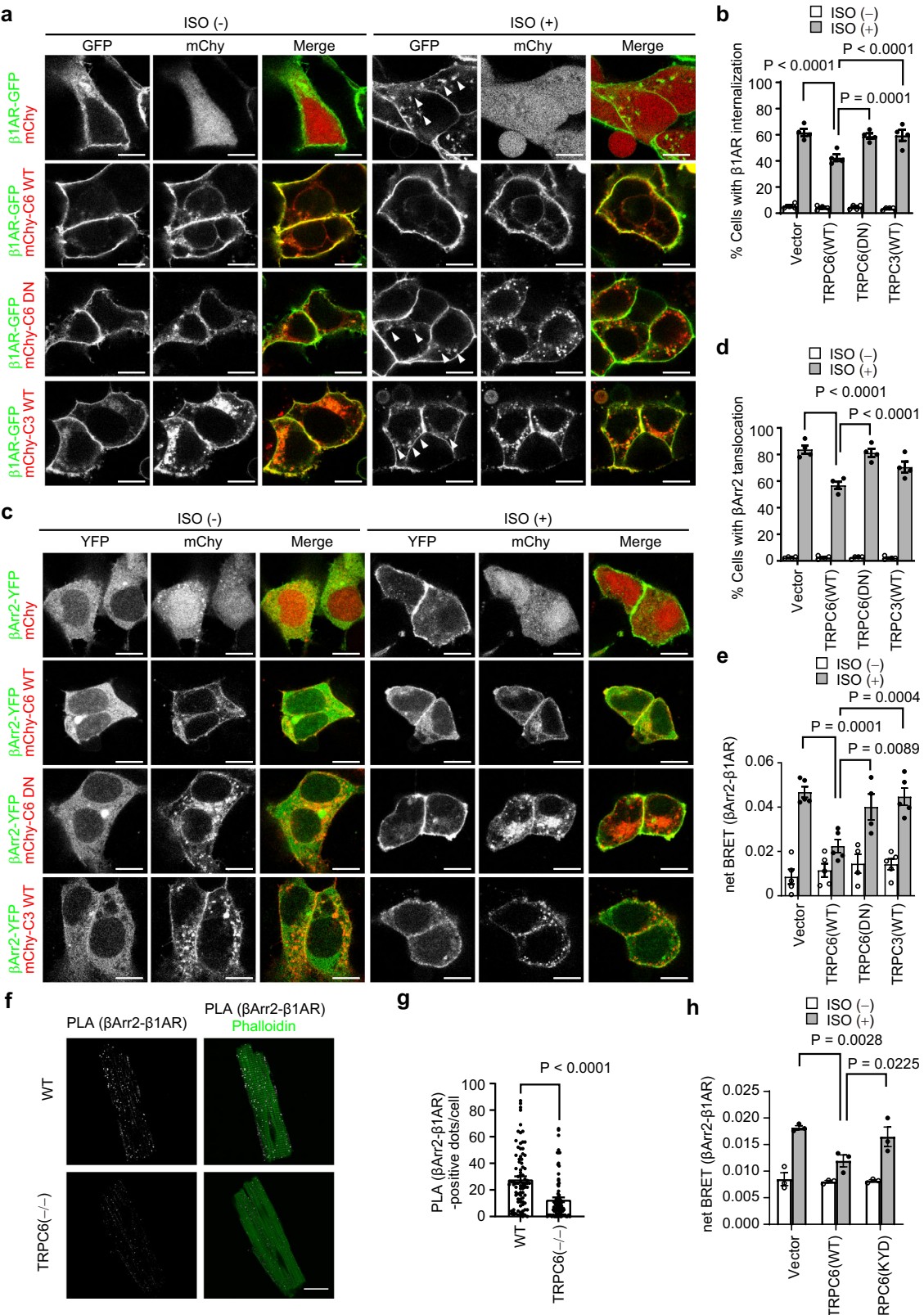

## STZ and MLP-KO models

For induction of hyperglycemia, male 129/Sv mice (8–9 weeks old) were injected intraperitoneally with STZ at a dose of 50 mg/kg body weight (dissolved in 0.1 M citrate buffer, pH 4.5) for 5 consecutive days. Control mice were injected with an equal volume of 0.1 M citrate buffer.

The basal cardiomyopathic phenotype of MLP-KO mice (129/Sv) was described previously[58,59]. A mini osmotic pump (Alzet) filled with PPZ2, a selective activator of TRPC6, or 50% (v/v) dimethyl sulfoxide (DMSO)/50% (v/v) polyethylene glycol 300 (vehicle) was implanted intraperitoneally into mice (8–9 weeks old). PPZ2 (2.5 mg/kg/day) or vehicle was continuously administered for 3 weeks.

**Fig. 5 | TRPC6 suppresses β₁AR internalization after ISO stimulation.**
**a, b** Representative images of HEK293 cells co-expressing β₁AR and TRPC6 or TRPC3 with or without ISO stimulation (**a**). HEK293 cells were transfected with β₁AR-GFP plasmid along with mCherry-tagged TRPC6 (mChy-C6 WT), pore-dead TRPC6 (mChy-C6 DN) or TRPC3 (mChy-C3 WT) vectors. Cells were treated with ISO (10 μM) for 30 min (stimulation condition is the same in **a**–**e**). Arrowheads indicate internalized β₁AR-GFP. Quantification of β₁AR-internalizing activity was determined as the percentage of the number of β₁AR-internalized cells to all cells (**b**). $n = 4$ each group. Scale bar, 10 μm. **c** and **d** Representative images of HEK293 cells co-expressing βArr2, β₁AR and TRPC6/3 with or without ISO stimulation (**c**). HEK293 cells were transfected with mCherry-tagged TRPC6, pore-dead TRPC6 or TRPC3 vectors along with YFP-tagged βArr2 (βArr2) and HA-β₁AR plasmids. Quantification of βArr2 translocation to the plasma membrane was determined as the ratio of the number of cells in which βArr2 was translocated after ISO stimulation to the

number of all cells (**d**). $n = 4$ each group. Scale bar, 10 μm. **e** Effects of TRPC6/3 on βArr2 recruitment to β₁AR. BRET signal between βArr2 and β₁AR. HEK293 cells were transfected with TRPC6, pore-dead TRPC6 or TRPC3 vectors with β₁AR-Rluc and βArr2-YFP plasmids. Cells were stimulated with ISO (10 μM) for 30 min. Vector, TRPC6 (WT), TRPC3 (WT), $n = 5$; TRPC6 (DN), $n = 4$. Representative images of PLA of βArr2 and β₁AR in adult cardiomyocytes isolated from WT or TRPC6$^{(-/-)}$ mice (129/sv). PLA signals were visualized as red spots counterstained with phalloidin (green) in merged images (**f**). Numbers of PLA signals were quantified in each cell (**g**). WT, $n = 84$ cells; TRPC6$^{(-/-)}$, $n = 82$ cells. Scale bar, 20 μm. **h** Effects of TRPC6(WT or KYD) on βArr2 recruitment to β₁AR. HEK293 cells expressing TRPC6(WT or KYD), β₁AR-Rluc, and βArr2-YFP were stimulated with ISO (10 μM) for 30 min. $n = 3$ each group. Data are shown as mean ± SEM. *$P < 0.05$; **$P < 0.01$ using two-way ANOVA followed by Sidak's post-hoc test (**b, d, e, h**) or the unpaired $t$-test (**g**, two-sided).

## TAC and MI operations

To induce cardiac pressure overload, TAC surgery was performed as described previously[15]. Briefly, WT and TRPC6$^{(-/-)}$ mice (8–12 weeks old) were anesthetized with isoflurane (FUJIFILM, Tokyo, Japan) and then intubated and ventilated. The chest cavity was opened at the intercostal area. The aorta was transversely constricted with a 7-0 nylon suture between the brachiocephalic artery and the left carotid artery to the width of a 27-gauge needle. After closing the chest cavity, buprenorphine was administrated intraperitoneally as an analgesic. A mini-osmotic pump (Alzet) filled with vehicle (50% (v/v) dimethyl sulfoxide (DMSO)/50% (v/v) polyethylene glycol 300) or PPZ2 was implanted intraperitoneally into mice at 7 days after TAC. PPZ2 (2.5 mg/kg/day) or vehicle was continuously administered for 4 weeks.

Surgery to induce MI was performed on 8-9-week-old male C57BL/6J mice as previously described[60]. All surgical procedures were performed in mice anesthetized with isoflurane. A mini-osmotic pump was implanted intraperitoneally into male C57BL/6J mice at 7 days after MI.

## Measurement of cardiac function

Hemodynamic parameters were measured using a micronanometer catheter (Millar 1.4F, SPR 671, Millar Instruments). Mice were anesthetized with isoflurane (Pfizer) at a concentration of 2.0–2.3% for induction. After insertion of the catheter to the left ventricular (LV) chamber, the isoflurane concentration was lowered to 1.5%–1.7% for maintenance. To maintain hemodynamic parameters constant before agonist infusion, measurements were started at least 10 min after the change to maintenance concentration. For analyzing responses to receptor stimulation in the autonomic nervous system, ISO (10 pg/g/min, 0.05 mL/h, intravenously (i.v.)) and ACh (0.5 mg/kg, 1.8 mL/h, i.v.) were injected to mice and LV parameters were measured. To measure hemodynamic parameters in blood vessels, a catheter was inserted to the aorta and mice were injected with NE (0.05 mg/kg, 1.8 mL/h, i.v.). For analyzing reflex responses, hydralazine (0.5 mg/kg, 1.2 mL/h, i.v.) was injected to mice. Compounds were injected to mice from the jugular vein using a cannula made of a polyethylene tube (internal diameter 0.28 mm) and syringe infusion pump (KDS100, KD Scientific). The average values of each parameter before injection were subtracted from values of each time point after injection; these calculated values are shown as Δ. The non-invasive echocardiographic measurement was performed using Nemio XG echocardiography (Toshiba) with a 14-MHz transducer for MLP-KO mice, Prospect T1 (Scintica) for TAC-operated mice and VEVO3100 (FUJIFILM) for MI-operated mice; measurements were performed under anesthesia with isoflurane (2.0–2.5% for induction).

## Isolation of adult mouse ventricular cardiomyocytes

Isolation of adult mouse ventricular cardiomyocytes was performed as described previously[61]. Briefly, a male mouse was heparinized (>100 units per mice, i.p.) and anesthetized (pentobarbital sodium, >300 mg/kg). The heart was quickly excised and the aorta was clamped with a

small vascular clamp. The heart was antegradely perfused with isolation buffer (containing (in mM) 130 NaCl, 5.4 KCl, 0.5 MgCl₂, 0.33 NaH₂PO₄, 25 HEPES, 22 glucose and 50 μU/mL bovine insulin (Sigma) (pH 7.4, adjusted with NaOH)) supplemented with 0.4 mM EGTA, followed by the perfusion of enzyme solution (isolation buffer containing 1 mg/mL collagenase type 2 (Worthington), 0.06 mg/mL trypsin (Sigma), 0.06 mg/mL protease (Sigma) and 0.3 mM CaCl₂). Thereafter, the LV chamber was removed and cut into small pieces in enzyme solution containing 0.2% bovine serum albumin (BSA) and 0.7 mM CaCl₂. The tissue-cell suspension was filtered through a cell strainer (100 μm, Falcon) and centrifuged for 3 min at $50 \times g$. After the supernatant was removed, the cell pellet was resuspended in isolation buffer supplemented with 0.2% BSA and 1.2 mM CaCl₂ and incubated for 10 min at 37 °C. After centrifugation ($50 \times g$, 3 min), the cells were resuspended in Tyrode's solution A (containing (in mM) 140 NaCl, 5.4 KCl, 1.8 CaCl₂, 0.5 MgCl₂, 0.33 NaH₂PO₄, 5.0 HEPES and 5.5 glucose (pH 7.4)) supplemented with 0.2% BSA. Cells were seeded to Matrigel (Corning)-coated 35 mm glass base dishes (glass 12Φ, IWAKI) or glass slides and incubated at 37 °C for 1 h. Cells were used within 1–6 h after isolation. For detubulation, ventricular cardiomyocytes were treated with 1.5 M formamide for 30 min[62].

## Isolation, culture and transfection of neonatal rat cardiomyocytes (NRCMs)

NRCMs were isolated from Sprague–Dawley rat pups on postnatal day 1–2. First, pups were sacrificed and ventricles were dissected and minced. The minced tissue was pre-digested in 0.05% trypsin-EDTA (Gibco) overnight at 4 °C and then digested in 1 mg/mL collagenase type 2 (Worthington) in PBS for 30 min at 37 °C with shaking (125 rpm). The dissociated cells were filtered through a cell strainer (70 μm, Falcon) and centrifuged for 2 min at $180 \times g$. After the removing supernatant, cell pellets were resuspended in Dulbecco's modified Eagle's medium (DMEM) supplemented with 10% FBS and 1% penicillin and streptomycin and plated in a 10-cm culture dish. Cells were incubated at 37 °C in a humidified atmosphere (5% CO₂, 95% air) for 90 min to separate non-cardiomyocytes. Floating NRCMs were collected and plated to Matrigel-coated culture dishes. After 24 h, the culture medium was changed to serum-free DMEM, and cells were incubated for more than 2 days before experiments.

For gene knockdown, the cells were transfected with siRNAs (20 nM) using Lipofectamine RNAiMAX (Invitrogen) and then incubated for 72 h. Stealth siRNAs for rat TRPC6 (#1, RSS331768; #2, RSS331769) and Stealth RNAi™ siRNA negative control (med GC) for siControl were obtained from Invitrogen. Cells were transfected with plasmid DNA using Lipofectamine 3000 (Invitrogen) and then incubated for 48 h.

## In vivo siRNA injection

Male C57BL/6J mice (8 weeks old) were injected with Stealth siRNA for mouse β₁AR (40 μg; MSS247004) or siControl (both from Invitrogen) encapsulated in HVJ-E (Ishihara Sangyo, LTD, Japan) via the orbital vein.

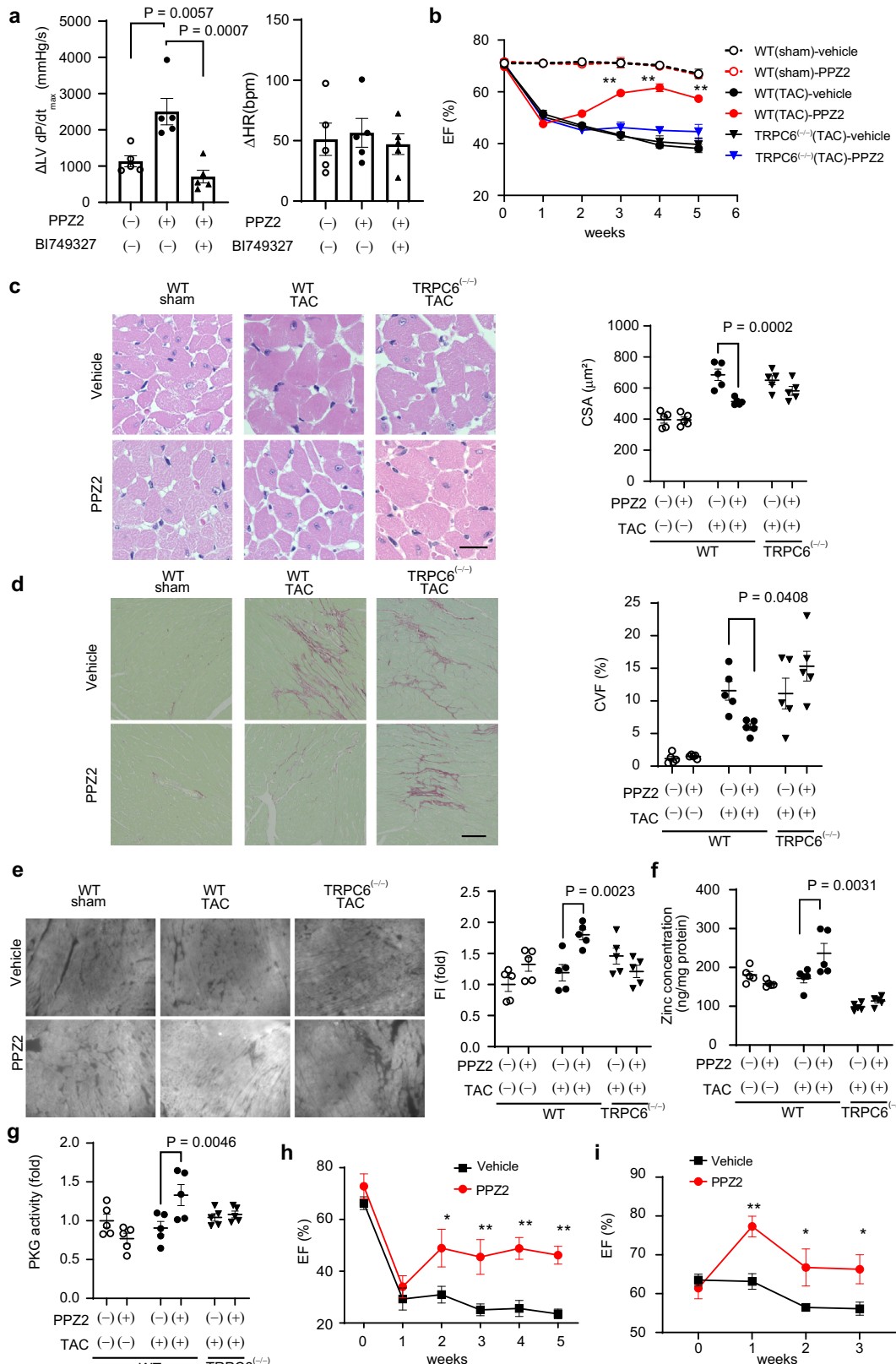

## HEK293 cell culture and transfection

HEK293 cells (ATCC Cat# TIB-71) were cultured in DMEM supplemented with 10% FBS and 1% penicillin and streptomycin. Cells were seeded at a density of $5 \times 10^5$ cells/dish in a 35 mm dish. On the following day, cells were transfected with plasmid DNAs using X-tremeGENE9 (Roche) and then incubated for 48 h. For imaging experiments, cells were replated on poly-L-lysine (Sigma)-coated glass coverslips or 35 mm glass base dishes (glass 12Φ) 1 day before experiments.

**Fig. 6 | Activation of the α1AR-TRPC6-βAR axis increases cardiomyocyte contraction. a** Effects of PPZ2 (2.5 mg/kg) and the TRPC6-selective inhibitor BI749327 (30 mg/kg). $n = 5$ each group. **b** Effect of PPZ2 (2.5 mg/kg/day) on the ejection fraction (EF) in TAC-operated WT and TRPC6$^{(-/-)}$ mice (129/sv). $n = 5$ each group; $P < 0.0001$ (3–5 weeks). **c** LV sections stained with hematoxylin and eosin. Scale bar, 25 μm. Average cross-sectional areas (CSAs) of cardiomyocytes. $n = 5$ each group. **d** LV sections stained with Sirius red (bars = 100 μm). Results of collagen volume fraction (CVF). $n = 5$ each. **e** Zn$^{2+}$ imaging of LV sections. Scale bar, 50 μm. $n = 5$ each group. **f** Zn$^{2+}$ concentration of heart tissue. $n = 5$ each group. **g** PKG activity. Effect of PPZ2 on EF in mice (C57BL/6J) after MI (**h**) and MLP-KO mice (129/sv) (**i**) ($n = 5$ each); $P = 0.0284$ (2 weeks), 0.0091 (3 weeks), 0.0023 (4 weeks), 0.0030 (5 weeks) in (**h**). $P = 0.0040$ (3 weeks), 0.0386 (4 weeks), 0.0386 (5 weeks) in (**i**). Data are shown as mean ± SEM. Significance was determined using one-way ANOVA with Tukey's comparison test (**a**) or two-way ANOVA followed by Sidak's comparison test (**b-h**) or Holm–Sidak's multiple comparisons post hoc test (**i**). *$P < 0.05$; **$P < 0.01$.

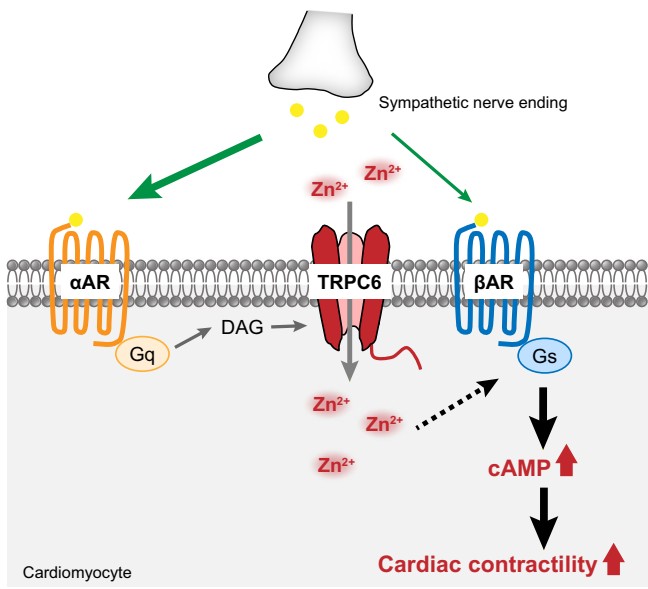

**Fig. 7 | Schema of the role of TRPC6-mediated Zn$^{2+}$ influx in adrenergic receptor-stimulated cardiac inotropy.** Cardiac inotropy through the sympathetic nervous system has been considered to be predominantly mediated by βAR activation, although the chemical binding affinity of NE to βAR is much lower than that to αAR. This study demonstrates that αAR-mediated Zn$^{2+}$ influx via TRPC6 promotes NE-induced cardiac inotropy by local Zn$^{2+}$-dependent enhancement of βAR-G$_s$ signaling. Downregulation of the αAR-TRPC6-Zn$^{2+}$ axis may reduce sympathetic nerve–mediated cardiac inotropy.

## Measurement of cardiomyocyte sarcomere length (SL) and calcium transient

The detailed method for SL measurement was described previously[63–65]. Briefly, isolated cells were stored in solution A containing (in mM) 128 NaCl, 2.6 KCl, 1.18 MgSO$_4$, 1.18 KH$_2$PO$_4$, 1.8 CaCl$_2$, 10 HEPES, 20 taurine and 11 glucose (pH 7.4). The myocytes were monitored through an inverted microscope (IX-70; Olympus) using a 40× long working distance objective lens (LUCPLN40×; Olympus). The cells were placed on a poly-HEMA (2-hydroxyethyl methacrylate; P3932, Sigma)-coated coverslip to prevent cell attachment to the bottom of the chamber. One of the cell ends was held by a short carbon fiber mounted in a glass capillary manipulated using a three-axis hydraulic manipulator (EDMS14-135a; Narishige) to prevent cells from being swept out of the field of view by the perfusate during the measurement. The cell was then perfused in Tyrode's solution B containing (in mM) 140 NaCl, 5.4 KCl, 1.8 CaCl$_2$, 1.0 MgCl$_2$, 5.0 HEPES and 11 glucose (pH 7.4) at 37 °C using an MPRE8 in-line heater (Cell Microcontrols) and electrically stimulated at 4 Hz using a MyoPacer cell stimulator (IonOptix Corporation). The myocyte SL changes were recorded at 240 Hz using IonOptix hardware and software (IonOptix Corporation). After recording stable SL traces in Tyrode's solution B without any drugs (control), the perfusion solution was changed to Tyrode's solution B containing ISO (10 nM) using a valve-controlled gravity perfusion system (VC3-4PG; ALA Scientific Instruments). The SL

recording was conducted until steady myocyte contraction was obtained in the presence of ISO.

For Ca$^{2+}$ transient measurements, cardiomyocytes were loaded with Fluo4-AM (2 μM) for 30 min at 37 °C after washing with Tyrode's solution A. Measurement of Ca$^{2+}$ transient was performed at 1 Hz pacing frequency with a pulse of 50 V, 10 ms duration using a pair of platinum wires placed on opposite sides of the chamber, which is connected to an electrical stimulator (SEN-3301, Nihon Kohden). Cells were stimulated with ISO (10 nM). Images were acquired at a rate of 5 Hz and analyzed using a video image analysis system (Metafluor, Molecular Devices).

## Zn$^{2+}$ imaging

Adult cardiomyocytes isolated from WT, TRPC6$^{(-/-)}$ and TRPC3$^{(-/-)}$ mice were seeded onto Matrigel-coated glass bottom dishes. Cells were loaded with the fluorescent Zn$^{2+}$ indicator FluoZin-3 (2 μM) for 30 min at 37 °C. Cells were treated with DTDP (100 μM) 2 min after starting the experiment. Images were acquired every 10 sec and analyzed using Metafluor.

For measurement of Zn$^{2+}$ influx, HEK293 cell were transfected with plasmid encoding α$_{1A}$AR and mCherry vector or TRPC6/3-mCherry. Cells were washed with HEPES-buffered saline solution (HBSS) containing (in mM) 140 NaCl, 5.6 KCl, 10 glucose, 10 HEPES 1 MgCl$_2$ and 1.8 CaCl$_2$ (at pH 7.4) and loaded with FluoZin-3 (2 μM) for 30 min at room temperature. After loading, the dye solution was replaced with HBSS. ZnCl$_2$ (50 μM) was applied 2 min after starting measurement. The α$_{1A}$AR agonist NE (10 μM) was added 3 min after ZnCl$_2$ application. Fluorescence images were acquired every 10 sec using a confocal microscope (TCS SP8 MP, Leica). For analysis, mCherry-positive cells were selected.

Splenocytes were isolated from the spleen in WT, TRPC3$^{(-/-)}$ and TRPC6$^{(-/-)}$ mice as described previously[66]. Primary vascular smooth muscle cells were isolated from mouse aorta in WT and TRPC6$^{(-/-)}$ mice at 5 week of age as previously described[67]. Cells were loaded with FluoZin-3 (2 μM) for 30 min at 37 °C. Cells were treated with DTDP (50 μM) 2 min after starting the experiment. Images were acquired every 10 sec and analyzed using Metafluor.

For measurement of Zn$^{2+}$ influx, TRPC6 (WT) and TRPC6 (KYD) stable rat aortic smooth muscle cells (RAOSMCs) were washed with HBSS and loaded with FluoZin-3 (2 μM) for 30 min at room temperature. After loading, the dye solution was replaced with HBSS. DTDP (50 μM) was applied 2 min after starting measurement. PPZ2 (30 μM) was applied 1 min after adding ZnCl$_2$ (50 μM). Fluorescence images were recorded and analyzed using Metafluor or a video image analysis system (Aquacosmos, Hamamatsu Photonics, Japan).

Cardiac tissue Zn$^{2+}$ imaging was performed as described in a previous study[68]. Briefly, a 1 mM stock solution of Zinpyr-1 (ZP-1) (Abcam, ab145349) was prepared in DMSO, and a 17 μM working solution was made using 0.9% saline. Heart slices (20-μm-thick) were covered in working ZP-1 solution for 2.5 min in a dark room and then washed with PBS to remove the ZP-1 solution. Images were obtained using a BZ-X800 fluorescence microscope (KEYENCE) and an 20× objective with consistent exposure time. Images were analyzed with ImageJ (NIH).

The $Zn^{2+}$ concentration in mouse hearts was measured using the Metallo Assay kit Zinc LS (Metallogenics Co., Chiba, Japan) following the manufacturer's protocol as described in a previous study[69].

## cAMP measurements

For fluorescence resonance energy transfer (FRET) imaging, NRCMs were seeded to Matrigel-coated glass base dishes on the day before siRNA transfection. Cells were transfected with siControl, siTRPC6-1 or siTRPC6-2 using Lipofectamine RNAimax. One day later, the cells were transfected with the Epac-based FRET sensor (Epac-S$^{H187}$; high-affinity version) using Lipofectamine 3000 and cultured for an additional 2 days before imaging. The biosensor was a gift from Prof. Kees Jalink[31]. Cells were washed by HBSS and stimulated with ISO (1 µM) 2 min after starting the experiment. Images were captured every 10 s using a confocal microscope (TCS SP8 MP). FRET was expressed as a ratio of CFP to YFP signals.

Intracellular cAMP levels were also measured using the cAMP EIA Kit (Cayman Chemical) following the manufacturer's instruction. NRCMs were seeded onto Matrigel-coated 12 or 24-well plates. After incubation in serum-free DMEM for 24–48 h or transfection of siRNA, cells were treated with prazosin (10 µM), propranolol (10 µM) and methoxamine (1 µM) for 24 h with extracellular $ZnCl_2$ (50 µM). Thereafter, cells were treated with 3-isobutyl-1-methylxanthine (IBMX; 500 µM, Sigma) for 1 h in the presence of each compound, following either NE (1 µM) or ISO (1 µM) stimulation for 30 min. In the TPEN-treated group, TPEN (25 µM) was added with IBMX. The control group was treated with DMSO. After stimulation of either NE or ISO, culture medium was aspirated and cells were lysed by adding 0.1 M HCl (200 µL for 12-well plate; 100 µL for 24-well plate) and incubated for 20 min at room temperature. The cell suspension was collected and centrifuged for 10 min at 1000 × g.

## Immunostaining and proximity ligation assay

Mouse heart tissues were embedded in optimal cutting temperature (OCT) compound (Sakura Finetek) and snap-frozen in a 2-methylbutane cooled by dry ice. Sections (12-µm-thick) were cut on a cryostat. Isolated adult mouse cardiomyocytes were plated on Matrigel-coated glass slides. Immunostaining was performed as follows: glass-mounted cryosection were fixed in 4% paraformaldehyde (PFA, dissolved in 1% PBS) for 7 min at room temperature. After rinsing with PBS, sections were blocked in PBS with 1% BSA, 0.3% Triton X-100 at room temperature for 1 h and then incubated with primary antibody against HCN4 overnight at 4 °C, following labeling by Alexa Fluor™ dye-conjugated secondary antibody and Alexa Fluor™ 488-conjugated wheat germ agglutinin (WGA; Thermo Fisher). Samples were mounted with Pro-Long™ Diamond Antifade Mountant with DAPI (Thermo Fisher).

To examine the TRPC6-$\beta_1$AR interaction and $\beta_1$AR-$\beta$-Arrestin 2 ($\beta$Arr2) interaction, proximity ligation assay was conducted using Duolink PLA Fluorescence (Sigma) following the manufacturer's instruction. After fixation and blocking, heart sections and cardiomyocytes were incubated with rabbit anti-$\beta_1$AR and either mouse anti-TRPC6 or anti-$\beta$Arr2 (Supplementary Table 9) for two nights at 4 °C, followed by incubation with probes for 1 h. The ligation (30 min) and amplification (150 min) steps were performed at 37 °C and samples were counterstained with Alexa Fluor™ 488–conjugated WGA (for heart sections) or Alexa Fluor™ 488-conjugated phalloidin (Thermo Fisher, for adult mouse cardiomyocytes). Nuclei were stained with DAPI. Images were captured using a confocal microscope (A1Rsi, Nikon) or BZ-X700 fluorescence microscope (Keyence). The number of PLA signal–positive puncta were quantified and analyzed using ImageJ (NIH) software.

## Visualization of $\beta$AR internalization and $\beta$Arr2 translocation

For visualization of $\beta$AR, HEK293 cells were transiently transfected with $\beta_1$AR-GFP plasmid along with mCherry, TRPC6(WT)-mCherry, pore-dead TRPC6 (TRPC6 DN)-mCherry or TRPC3-mCherry vectors. For visualization of $\beta$Arr2 translocation, HEK293 cells were transfected with hemagglutinin-tagged $\beta_1$AR ($\beta_1$AR-HA) and $\beta$Arr2-YFP plasmids along with mCherry, TRPC6(WT)-mCherry, TRPC6(DN)-mCherry or TRPC3-mCherry vectors. Two days after transfection, cells were treated with ISO (10 µM) for 30 min. Images were captured using confocal microscopy (A1Rsi).

## Bioluminescence resonance energy transfer (BRET)

HEK293 cells were transfected with $\beta_1$AR-Rluc, $\beta$Arr2-YFP or $G\alpha_s$-YFP plasmid Readings along with pCI-neo, TRPC6(WT), TRPC3(WT) or TRPC6(DN) vectors. At 48 h after transfection, cells were detached with PBS/EDTA (2 mM) and centrifuged for 5 min (1000 × g) at room temperature. After centrifugation, cell pellets were resuspended in BRET buffer (1.8 mM $CaCl_2$, 1 mM $MgCl_2$, 0.1% glucose in PBS) and moved to 96-well white microplates. Cells were stimulated with ISO (10 µM) for 30 min and incubated with coelenterazine h (5 µM, Wako) for 10 min. The BRET signal was determined as the ratio of the light emitted by YFP and the light emitted by Luc using a fluorescent microplate reader (SpectraMax® i3, Molecular Devices). The values were corrected by subtracting the background BRET signals.

## Plasmid construction

The PDZ3 domain of mouse PSD-95 and the full-length mouse Snapin were linked through a 116 amino acid flexible EV linker[70]. Flag-tag was fused to the N-terminus of the protein, and the t2a-mCherry sequence was fused to the C-terminus of the protein. This construct was cloned into the pCDNA3 vector and verified by sequencing.

Detailed information of pEGFP-N1 vector encoding $\beta_1$AR, $\beta_1$AR-Rluc or AT1R-Rluc, pcDNA3.1 vector encoding $\beta$Arr2-YFP or $\alpha_{1A}$AR, pCI-neo vector encoding FLAG-TRPC6, TRPC6, TRPC6(DN) or TRPC3, and pmCherry-N1 vector encoding TRPC6, TRPC6(DN) or TRPC3 was described in the previous study[11,51]. Point mutations of plasmids were generated by site-directed polymerase chain reaction (PCR) mutagenesis. $G\alpha$s-YFP plasmid was a gift from Catherine Berlot (Addgene plasmid #55781; http://n2t.net/addgene:55781; RRID: Addgene_55781)[71].

## Analysis of contractility

NRCMs ($2 \times 10^4$ cells/well) were seeded into 96-well plates. Microscopy images of NRCMs were recorded for 10 seconds; imaging was performed at 14.5 frames per second on a BZ-X800 microscope (Keyence). Each video was analyzed using ImageJ (NIH), as previously described[72].

## Histological analysis of mouse hearts

The paraffin-embedded heart sections (5 µm in thickness) were stained with hematoxylin and eosin (H&E) or Sirius red, and the cell-sectional area (CSA) of cardiomyocytes and collagen volume fraction (CVF) of heart sections were analyzed using the BZ-X800 analyzer (Keyence) and Image J[59].

## PKG activity

PKG activity was assessed by EIA colorimetric assay (CycLex) as described previously[44]. Briefly, kinase reaction buffer containing 2.5 mM ATP and 500 µM cGMP was prepared following the manufacturer's instructions and added to protein lysates prepared from myocardial tissue in a 96-well reaction plate; samples were incubated at 30 °C for 20 min. After washing, HRP-conjugated anti-phospho-specific antibody was added to the wells for 60 min at 22 °C. Samples were washed and then substrate reagent was added to the wells for 5–15 min; the reaction was halted using stop solution, and absorbance at 450/550 nm was measured using the Nivo microplate reader (PerkinElmer, Waltham, MA, USA). The cGK positive control was obtained from CycLex and used following the manufacturer's instructions.

### Radioligand binding assay

Mouse heart tissues were homogenized in homogenate buffer (25 mM Tris-HCl (at pH 7.4) and 2 mM EDTA) using a Physcotron homogenizer. The homogenate was centrifuged at $800 \times g$ for 10 min at 4 °C. The supernatant was ultracentrifuged at $125,000 \times g$ for 30 min at 4 °C. The pellet was resuspended in assay buffer (75 mM Tris-HCl (at pH 7.4), 2 mM EDTA and 12.5 mM $MgCl_2$) using a Dounce homogenizer. For the receptor binding assay, 50 µg protein in a total volume of 200 µL of assay buffer with or without 20 µM propranolol were added to tubes. Samples were incubated with the radiolabeled antagonist $[^{125}I]CYP$ (final concentration 12.5–400 pM) for 90 min at 30 °C. The reactions were stopped by the addition of an ice-cold assay buffer and filtered through Whatman GF/C glass microfiber filters presoaked in 0.1% polyethylenimine. The filters were washed three times with ice-cold assay buffer and air-dried. The radioactivity was measured using a liquid scintillation counter.

### RNA isolation and real-time polymerase chain reaction (RT-PCR)

Total RNA was isolated from frozen mouse hearts using the RNeasy Fibrous Tissue Mini Kit (Qiagen) following the manufacturer's instruction. Complementary DNA was synthesized with ReverTra Ace® qPCR RT Master Mix (TOYOBO). RT-PCR was performed using Lightcycler® 96 (Roche) with the KAPA SYBR® FAST qPCR kit (KAPA BIOSYSTEMS) following the manufacturer's instructions. The primer sequences used for RT-PCR are listed in Supplementary Table 10.

### Measurement of biochemical parameters

In the STZ model, non-fasted blood glucose levels were determined by Glucose Pilot (Technicon) 4 weeks after STZ injection.

### Microarray analysis

Microarray analysis was performed as described previously[73]. Briefly, RNA samples were converted into biotinylated cRNA using Two-Cycle Target Labeling and Control Reagents (Thermo Fisher Scientific). Labeled RNA was processed for microarray hybridization to a GeneChip Mouse Genome 430 2.0 array (Thermo Fisher). An Affymetrix GeneChip Fluidics Station was used to perform streptavidin/phycoerythrin staining. The hybridization signals were scanned using a GeneChip Scanner 3000 (Thermo Fisher) and analysis of array data was performed with Expression console software 1.2 (Thermo Fisher). Normalization was performed with MAS5 algorithm using a set of mouse maintenance genes (Thermo Fisher). These data have been deposited in NCBI's Gene Expression Omnibus (GEO) and are accessible through GEO Series accession number GSE189494.

### Western blot

Frozen heart samples were lysed in lysis buffer containing (in mM) 120 NaCl, 20 Tris-HCl, 1%(v/v) TritonX-100, 0.1% SDS, 0.5% sodium deoxycholate, 10% glycerol, 1 EDTA (pH 7.4) and protease inhibitor cocktail (Nacalai). Cell lysates were sonicated and clarified by centrifugation for 10 min at $10,000 \times g$ at 4 °C. Protein concentrations were determined (DC Protein Assay, Biorad) and equal amounts of proteins were suspended in SDS sample buffer containing 0.1 M DTT and incubated for 1 h at room temperature. Protein samples were separated by SDS-PAGE and transferred onto PVDF membranes (Millipore). The membranes were incubated with the indicated primary antibodies overnight at 4 °C, followed by incubation with secondary antibodies. Immunoreactive bands were detected using Western Lightning Plus ECL (PerkinElmer) and images were captured with an ImageQuant LAS 4000 (GE Healthcare Life Science).

### Immunoprecipitation assay

HEK293 cells expressing FLAG-tagged TRPC6 and EGFP-fused $\beta_1AR$ ($\beta_1AR$-EGFP) were lysed in lysis buffer containing 1% Triton X-100, 140 mM NaCl, 1 mM EDTA, 20 mM Tris-HCl (pH 7.4) and protease inhibitor cocktail (Nacalai, Japan). Cell lysates were incubated with FLAG M2 agarose (Sigma, USA) for 12 h to immunoprecipitate FLAG-TRPC6. Immune complexes were washed three times with lysis buffer and eluted with 0.1 mg/ml FLAG peptide (Sigma, USA) in lysis buffer. The immunoprecipitated proteins were assayed by western blotting using antibodies (Supplementary Table 9).

### Whole cell patch clamp

TRPC6 channel currents were measured using the whole-cell patch-clamp technique with an EPC-10 patch-clamp amplifier (Heka Elektronik). Patch electrodes with a resistance of 3–4 MΩ (when filled with internal solution) were made from 1.5-mm borosilicate glass capillaries (Sutter Instrument). Voltage ramps (−100 to +100 mV) of 250 ms were recorded every 2 s from a holding potential of −60 mV, and the currents were normalized on the basis of cell capacitance. Cells were allowed to settle in the perfusion chamber in the external solution containing (in mM) 140 NaCl, 5.6 KCl, 0.8 $MgCl_2$, 1.8 $CaCl_2$, 10 HEPES and 10 glucose (pH 7.4). For the analysis of TRPC6 channel activity in the presence of $Zn^{2+}$, $CaCl_2$ was replaced with 10 mM $ZnCl_2$. The pipette solution contained (in mM) 120 CsOH, 120 aspartate, 20 CsCl, 2 $MgCl_2$, 5 EGTA, 1.5 $CaCl_2$, 10 HEPES and 10 glucose (pH 7.2, adjusted with Tris base). Cells were superfused with standard external solution in the presence or absence of carbachol focally using a Y-tube perfusion system.

### Selection of specific cell/animal models

Considering the statistical power for multiple comparisons of in vivo experiments, we used mice that can breed in large numbers at the same time and that are easy to genetically modify. To exclude the possibility of artifacts from strain differences, the pharmacological effect of PPZ2 was evaluated using C57BL6J background mice (for MI) in addition to 129/Sv mice. Acutely isolated mouse cardiomyocytes are difficult to maintain in long-term culture, and therefore culture experiments were mainly performed using high-isolation-yield rat cardiomyocytes. HEK293 and RAOSMC cells are easy-to-use cell lines for transient expression of multiple proteins simultaneously and stable expression of TRPC6, respectively. The TRPC6-mediated $Zn^{2+}$ influx activity was measured using subculture-able cells isolated from TRPC6$^{(−/−)}$ mice, such as splenocytes and vascular smooth muscle cells, to confirm the universality of the $Zn^{2+}$ pool controlled by TRPC6.

### Statistical analysis

G*Power3.1.9.2 software was used to calculate the sample size for each group (RRID:SCR_013726). All results are presented as the mean ± SEM from at least three independent experiments. Statistical comparisons were carried out by unpaired t tests for two-group comparisons, one-way analysis of variance (ANOVA) followed by Tukey's *post hoc* test or two-way ANOVA followed by Sidak's comparison test for multiple groups. $P < 0.05$ was considered statistically significant. Statistical analysis was performed using GraphPad Prism 9.0 (GraphPad Software, LaJolla, CA).

### Reporting summary

Further information on research design is available in the Nature Research Reporting Summary linked to this article.

## Data availability

The data supporting the findings from this study are available within the article and its Supplementary information. The analyzed microarray data have been deposited in NCBI's Gene Expression Omnibus (GEO) and are accessible through GEO accession number GSE189494. Any remaining raw data will be available from the corresponding authors upon reasonable request. Source data are provided with this paper.

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

## Acknowledgements

We thank Dr. Mi Xinya and Yuya Taniguchi for support with cardiomyocyte isolation and recombinant AAV construction. We also thank the Spectrography and Bioimaging Facility (NIBB Core Research Facilities). We thank Mitchell Arico and Gabrielle White Wolf, Ph.D., from Edanz (https://jp.edanz.com/ac) for editing a draft of this manuscript. This work was supported by JST CREST Grant Number JPMJCR2024 (20348438), JSPS KAKENHI (Grant Numbers JP20H05512, JP20H03673, JP21H05269, JP22H02772, JP22K19395, JP21K15338, and JP22H04814), and AMED (Grant Number JP22ama121031 and JP21ek0109509). This work was supported in part by the National Research Foundation of Korea (NRF) funded by the Korean government (MSIP) (2017K1A1A2004511) and a grant of Planned Collaborative Project from the National Institute for Physiological Sciences (NIPS; 21-212).

## Author contributions

S.O., K.N., and M.N. designed the study and wrote the manuscript; S.O., K.N., Y.F., Y.Y., A.N., X.T., R.O., S.M., Y.K., T.N-T., T. Ka., T.Ku. and Y.S. performed experiments and analyzed and interpreted data; M.H. and M.S. generated TRPC6(KYD/KYD) mice; Y.N., K.K., R.N., Y.S., Y.M. and G.I. contributed reagents/analytic tools and provided critical suggestions; M.N. edited the manuscript.

## Competing interests

The authors declare no competing interests.

## Additional information

[1]National Institute for Physiological Sciences (NIPS), National Institutes of Natural Sciences, Okazaki 444-8787, Japan. [2]Exploratory Research Center on Life and Living Systems (ExCELLS), National Institutes of Natural Sciences, Okazaki 444-8787, Japan. [3]Department of Physiological Sciences, SOKENDAI (School of Life Science, The Graduate University for Advanced Studies), Aichi 444-8787, Japan. [4]Graduate School of Pharmaceutical Sciences, Kyushu University, Fukuoka 812-8582, Japan. [5]Asahikawa Medical University, Hokkaido 078-8510, Japan. [6]Shinshu University School of Medicine, Matsumoto 390-8621, Japan. [7]Faculty of Science, Mahidol University, Bangkok 10400, Thailand. [8]National Institute of Health Sciences, Kanagawa 210-9501, Japan. [9]Kyoto University Graduate School of Medicine, Kyoto 606-8507, Japan. [10]Graduate School of Pharmaceutical Sciences, Osaka University, Osaka 565-0871, Japan. [11]Graduate School of Engineering, Kyoto University, Kyoto 615-8510, Japan. [12]These authors contributed equally: Sayaka Oda, Kazuhiro Nishiyama.
✉e-mail: nishida@phar.kyushu-u.ac.jp

