## [Peer Review File · Nature Communications]

Myocardial TRPC6-mediated Zn²⁺ influx induces beneficial positive inotropy through β -adrenoceptorsREVIEWER COMMENTS

Reviewer #1 (Remarks to the Author):

Oda et al describe an involvement of the TRPC6 channel in baroreceptor mediated positive inotropy, which they describe as beneficial positive inotropy.

The authors describe a complex series of signaling components leading to an increased cardiac contractility that may help to explain two important phenomena occurring in heart failure: The decompensation of the heart after α 1-blocker therapy (ALLHAT) and the hypercontraction of the base region (vs. ballooning in the apical region) in Takotsubo cardiomyopathy. Even though several of these signaling components such as the permeability of TRPC6 for Zn^{2+} , the fact that it can contribute to acute cardiac contractility (Seo et al., 2014), its activation by the sympathetic nervous system (Inoue et al., 2001) have been described before, the interconnection seems new.

The design of the study is elegant and contains many controls. However, there are many open questions and additional controls are needed, especially – as the authors cite – because many contrary data have been reported. To be convincing the study needs strong and extensive additional data.

Major comments:

Heart Failure model: The authors propose the enhanced Zn^{2+} influx with α 1AR stimulation as a new therapeutic target for chronic heart failure. As mentioned above, it would be a very interesting explanation why α 1-blockers initiate cardiac decompensation in some patients. However, the data presented in the manuscript with regards to the outcome “beneficial or maladaptive” is vague. A) The authors are using a rather short-term model of three weeks, that does not even develop heart failure (Fig. 6d/e; no significant decrease in EF or FS). B) The n-number used is extremely low and C) the MLP-KO mouse model is a model of dilated cardiomyopathy, but it is not typical of clinical heart failure, and there are more clinically relevant models. Why was this model chosen? If the authors want to convince that TRPC6 can induce beneficial long-term effects, they need to use models that have been used before for related studies such as myocardial infarction or TAC. D) Moreover, the compound PPZ2 used to validate their hypothesis is unspecific and also targets TRPC3 and TRPC7 (Sawamura et al., 2016)(also off-target effects of PPZ2 should be addressed; this should be possible in parts in TRPC6-KO mice), so that the key conclusion of the manuscript about the long-term beneficial effects in heart failure due to TRPC6-mediated Zn^{2+} influx certainly cannot be drawn by this experiment. E) The authors show nicely the short-term effects of TRPC6 in vitro and in cardiac catheterization experiments, but thus far there is no explanation given for the anti-hypertrophic and anti-fibrotic effect they propose in the “heart failure” model.

In their previous Sci Rep paper, the authors speculate that the background of the mice may contribute to the differential outcome of the TRPC6-KO (Oda et al., 2017): “Using TRPC6-deficient 129/Sv background mice, we found that TRPC6 deletion failed to suppress pressure overload-induced LV dysfunction as well as oxidative modification of plasma Gpx3 protein, despite significant

suppression of interstitial fibrosis.” If this is the case, comparative studies need to be done to clarify the potential impact of TRPC6 in heart failure. The background of the MLP-KO is not given.

In Fig. 2, a close proximity of b1AR and TRPC6 is proposed to explain the specificity of the proposed signaling pathway. Even though the pictures look nice, controls are missing for antibody specificities. bAR antibodies are rather problematic and especially the detection of endogenous bAR is rather difficult. Convincing antibody controls and possibly PLA assays of the overexpressed receptors with respective Taqs could contribute to the validation of the experiments, also Western blots for the base and apical region of the heart. Is the close b1AR and TRPC6 proximity specific for b1AR? Other interaction assays would help to substantiate this hypothesis.

The authors show that ISO-induced inotropy and chronotropy are similarly affected in TRPC6-KO and TRPC3-KO (Fig.1c/d) as in response to baroreceptor-induced inotropy (Fig.1a/b). Same for Fig. 3, i.e. ISO-induced sarcomere contraction/Ca²⁺ cycling. This is somewhat puzzling since the authors – at least that is my impression – do not propose an impact of basal TRPC6 physiological function. How can then the TRPC6-KO impact on ISO-stimulation if bAR are downstream of TRPC6 (also see scheme in Fig.7)? Can PPZ2 enhance the ISO-mediated effects? Is the TRPC6-phenotype unaltered if PPZ2 is applied? Are these effects in Fig1 and Fig3 really due to Zn²⁺ influx but not Ca²⁺ influx? If the low levels of intracellular Zn²⁺ are thought to contribute to the contractile phenotype in response to ISO, it should be analysed whether the TRPC6-KO cells with long-term low Zn²⁺ level do not possibly suffer from ER stress or other dysfunctional proteins that may impact on ISO-stimulated Ca²⁺ cycling and sarcomere contraction?

Increased intracellular Zn²⁺ levels/dysregulated Zn²⁺ homeostasis have been associated with increased ER stress and RyR2 leakiness in cardiomyocytes and are hypothesized to contribute to the development of heart failure (e.g. Reilly-O'Donnell et al., JBC, 2017 or Olgar et al., 2018). ER stress should be analyzed in the TRPC6-KO and the long-term mouse model.

In Fig.4, it is shown that the beta-blocker propranolol antagonizes the a1AR-TRPC6-Zn²⁺ axis. If this is the case, is TRPC6 a promising therapeutic target in heart failure since beta-blockers are given to most heart failure patients? This should be discussed at least. Further, PKG signaling is thought to be beneficial in heart failure, and PKG has also been shown to impact on TRPC6 and thereby to attenuate excess SSC and arrhythmia (Seo et al., Circ Res 2014).

In Fig. 2, a close proximity of b1AR and TRPC6 is proposed and it is also speculated that TRPC6 interacts with a1AR in cardiomyocytes and that this contributes to the proposed mechanism (a1AR-TRPC6-Zn²⁺-b1AR). Is this a triple complex that is postulated? And why should TRPC6 not lead to a Ca²⁺ influx in cells that do not express b1AR (Suppl Fig. 5b/c)? Does the complex contribute to the specificity of TRPC6 to certain ions? These hypotheses need to be clarified. The authors state that TRPC6 does not impact on bArr recruitment to the AT1R, but can the AT1R as Gq coupled receptor also increase the responsiveness of bAR via TRPC6? Is it the proximity or snapin that leads to the proposed specificity of a1AR for TRPC6?

Further comments:

o Table 3:

- In table 3 the data of the echos after 1 and 2 weeks are missing. They need to be included.
- The n-numbers are missing in the table. In case the n-number is “3”, the number is extremely low.
- The data in table 3 rather suggest (difficult with n=3) an increase of cardiac hypertrophy by PP2Z (Table 3; IVS 0.69 vs 0.81 and LVPW 0.72 vs 1.03), while the CSA (Fig,6f) shows a significantly reduced cardiomyocyte size. How can this be explained? Could a concomitant calcium influx be responsible for that?
- The type of mouse used needs to be mentioned in the legend.

o n-numbers should be given in all legends.

o Quality of blots in Suppl Fig. 3 is low, especially for GRK2. This needs to be reduced and a control would be of help.

Reviewer #2 (Remarks to the Author):

In this manuscript, the authors revealed the physiological role of the TRPC6 channel in bAR-dependent positive inotropy in mouse hearts and described the novel mechanism on TRPC6-mediated Zn²⁺ influx in cardiomyocytes. The previous studies from the same group revealed that TRPC3 channels contribute to oxidative stress-related heart failure through a protein-protein interaction with NADPH oxidase, but TRPC6-deficient mice show the opposite effect. Therefore, the current ms focused on the specific role of TRPC6 channels with parallel measurements of TRPC3 expression and activity. This is a very detailed study, and the results of the study support the authors' conclusions. The experimental design and methods used for this study are sound. There are several suggestions that could improve overall excellent and novel study.

1) There are several small molecules specific for TRPC6, which work in vivo. As an example, SAR-7334 (1), laryxil acetate (2), and BI 749327 (3) are active in the nanomolar range and have more than an order of magnitude of selectivity for TRPC6 over the closely related TRPC3 and TRPC7 channels and do not block more distantly related TRPC channels, including TRPC4 and TRPC5. Use any of these compounds in vivo should greatly benefit the current study and provide direct evidence of TRPC6 contribution.

2) The authors clearly describe the TRPC6-mediated Zn²⁺ influx, but potential calcium influx through the same mechanism should be better addressed. The authors should test if the exact mechanism affects TRPC6-mediated Ca²⁺ influx in the same cells, which might also improve chronic heart failure in mice.

3) M085 and GSK1702934A, two small molecules, which directly activate TRPC6 through a mechanism of stimulation of extracellular sites formed by the pore helix and transmembrane helix (4). It would be great if the authors test one of these agonists on Zn²⁺ influx for both wild-type and mutated channels.

4) Some experiments have only n=3 (f.i., data reported in Figure 6), which appears a small number of experiments.

5) Multiple cells lines and rodents (both mice and rats) were utilized for these experiments. The authors should provide some brief rationale for the selection of specific cell/animal models (technical limitations, etc).

6) Data shown in supplementary figures 4a-4e could be added to the main figures if space allows.

7) Recent study about zinc as a target for vascular therapeutics could be discussed (Betrie et al., Nat Commun, 2021; PMID: 34075043).

1. Maier T, Follmann M, Hessler G, Kleemann HW, Hachtel S, Fuchs B, et al. Discovery and pharmacological characterization of a novel potent inhibitor of diacylglycerol-sensitive TRPC cation channels. *Br J Pharmacol*. 2015;172(14):3650-60.

2. Urban N, Wang L, Kwiek S, Rademann J, Kuebler WM, Schaefer M. Identification and Validation of Larixyl Acetate as a Potent TRPC6 Inhibitor. *Mol Pharmacol*. 2016;89(1):197-213.

3. Lin BL, Matera D, Doerner JF, Zheng N, Del Camino D, Mishra S, et al. In vivo selective inhibition of TRPC6 by antagonist BI 749327 ameliorates fibrosis and dysfunction in cardiac and renal disease. *Proc Natl Acad Sci U S A*. 2019.

4. Yang PL, Li XH, Wang J, Ma XF, Zhou BY, Jiao YF, et al. GSK1702934A and M085 directly activate TRPC6 via a mechanism of stimulating the extracellular cavity formed by the pore helix and transmembrane helix S6. *J Biol Chem*. 2021;297(4):101125.

Reviewer #3 (Remarks to the Author):

This is a very nice and interesting study showing the pivotal role of TRPC6 channels in modulating the effect of alpha1-adrenoceptors on cardiac inotropy. It solves the long-standing question and mechanistically explains why these receptors seemingly do not directly contribute to contractility regulation in adult myocardium but have an important effect on heart function in the context of cardiac remodelling. The authors demonstrate that it is specifically the TRPC6 but not TRPC3 channels which is regulated downstream of alpha receptors and is responsible for Zn ion influx which in turn modulated beta-adrenoceptor signalling in terms of G protein coupling in activation of cAMP production. Although the comprehensive and extensive results of the manuscript greatly support authors' conclusions there are some experimental issues and minor lacks in mechanistic insight which can be improved prior to publication:

1. Figure 1a,b nicely shows that the deletion of TRPC6 reduces the positive inotropic (but not chronotropic) effect of Hydralazine. How about TRPC3 knockout under these conditions?

2. Figure 2 shows nicely that TRPC6 co-localizes with beta1-AR in LV but not SAN cells. Can the authors comment in the discussion on the possible localisation of TRPC6 to submembrane structures of myocytes? As far as I remember there was some literature that TRPCs can be found in T-tubules of myocytes. If the co-localisation of beta1-AR takes place there, these could explain why SAN cells (which have less T-tubules per se) do not show this phenomenon. If the authors want to go in more detail here experimentally, one could check if chemical detubulation of myocytes or cholesterol depletion would reduce PLA signal.

3. Having said that it is unfortunate that cAMP FRET measurements were performed in neonatal instead of adult myocytes since neonatal cells do not have a mature well organised membrane structure including T-tubules. I will highly appreciate if such measurements could be done in adult cells, though I understand that an adenovirus would be required for transduction of these cells with the sensor. If this is not technically possible for the lab, maybe they could do cAMP ELISA on adult WT vs KO myocytes. It is also not well discussed in the text how the authors explain the reduced cAMP response to ISO in TRPC6 KO cell in Fig 3e,f. ISO is a beta-selective agonist which does not activate alpha-receptors. How does the alpha-AR-TRPC6-Zn connection feed into here or it is just the "basal" TRPC6 activity which matters in this case?

Calcium measurements in Fig 3 show data analysis only on the peak height which is inotropic response, did the authors analyse also the rate of Ca or relaxation decay, in other words does TRPC6 affect the positive lusitropic effect of ISO and/or cell relaxation at basal state?

4. The choice of MLP mouse models as a dilated hypertrophy model is not well justified. The authors should provide a clear rationale why this mouse model was chosen for the study. Will the protective

effect of the peptide be present also in other classical heart failure model such as after pressure overload or myocardial infarction?

5. Reference 26 appears two times to quote different studies. The first one referring to Takotsumo cardiomyopathy might be Paur H et al. Circulation 2012?

6. Mechanistically the connection between Zn and beta1-AR/Gs/cAMP pathway is referred to largely based on previously published literature showing that Zn can effect Gs, GRK activities and receptor affinity for ligands such as ISO or NE, no experiments on it were done for this particular paper. I feel

7. Figure 6a,b - there is a discrepancy between positive inotropic response to NE in control vs RFP expressing cells (3.5 vs 2 fold increase), and almost no effect of NE in Fig 6c control column. Can the authors explain this discrepancy?

8. Fig. 4d - cells were pretreated with receptor blockers for 24 or 25 h. Why that long? The blockers should work normally almost instantaneously.

Responses to Reviewers

We truly appreciate your time and effort in reviewing our manuscript. We carefully reviewed each of the comments and have provided additional text in the manuscript as well as performed additional experiments. Our response to each point is presented below. Modifications to the text in the manuscript are indicated by red font. We hope that our responses have addressed the concerns raised by the reviewers.

Responses to Reviewer 1.

Major comments:

Heart Failure model: The authors propose the enhanced Zn^{2+} influx with $\alpha 1AR$ stimulation as a new therapeutic target for chronic heart failure. As mentioned above, it would be a very interesting explanation why $\alpha 1$ -blockers initiate cardiac decompensation in some patients. However, the data presented in the manuscript with regards to the outcome “beneficial or maladaptive” is vague. A) The authors are using a rather short-term model of three weeks, that does not even develop heart failure (Fig. 6d/e; no significant decrease in EF or FS). B) The n-number used is extremely low and C) the MLP-KO mouse model is a model of dilated cardiomyopathy, but it is not typical of clinical heart failure, and there are more clinically relevant models. Why was this model chosen? If the authors want to convince that TRPC6 can induce beneficial long-term effects, they need to use models that have been used before for related studies such as myocardial infarction or TAC. D) Moreover, the compound PPZ2 used to validate their hypothesis is unspecific and also targets TRPC3 and TRPC7 (Sawamura et al., 2016)(also off-target effects of PPZ2 should be addressed; this should be possible in parts in TRPC6-KO mice), so that the key conclusion of the manuscript about the long-term beneficial effects in heart failure due to TRPC6-mediated Zn^{2+} influx certainly cannot be drawn by this experiment. E) The authors show nicely the short-term effects of TRPC6 in vitro and in cardiac catheterization experiments, but thus far there is no explanation given for the anti-hypertrophic and anti-fibrotic effect they propose in the “heart failure” model.

[Response]

Thank you very much for the helpful comments. To address the issue regarding the mouse models, we performed additional experiments and have included data demonstrating the cardioprotective effects of PPZ2 using three different heart failure model mice: the transverse aortic constriction (TAC), myocardial infarction (MI), and muscle LIM protein (MLP)-deficient dilated cardiomyopathy. All model mice were monitored for 6 weeks. Treatment of TAC-operated mice with PPZ2 after development of LV dysfunction significantly improved cardiac contractility, cardiomyocyte hypertrophy, and fibrosis, and the cardioprotective effect of PPZ2 was abolished in

TRPC6-deficient TAC mice (Fig. 6a–d and Supplementary Table 3, 4). PPZ2 treatment also increased intracellular Zn^{2+} and PKG activity in TAC-operated mouse hearts (Fig. 6e–g). PPZ2 also improved cardiac contractility, cardiomyocyte hypertrophy, and fibrosis in MLP-deficient dilated cardiomyopathic mice and mice with chronic heart failure 6 weeks after myocardial infarction (Fig. 6h, i and Supplementary Fig. 8a–f and Supplementary Table 5-8). As indicated and to confirm our results, PPZ2 experiments in the heart failure models were also all performed in TRPC6-KO mice. Based on our observations, we believe that maintaining the cardiac positive inotropy through TRPC6-mediated Zn^{2+} influx by PPZ2 treatment partially contributes to the prevention of LV remodeling after MI and TAC.

In addition, we newly found that PPZ2 treatment also increased the activity of protein kinase G (PKG), an anti-hypertrophic factor, in a TRPC6-dependent manner (Fig. 6g). The results that the increases in Zn^{2+} level and PKG activity were observed only in TAC-operated hearts are plausible because TRPC6 is upregulated in TAC-mouse hearts. β AR stimulation is reported to activate the NOS3-dependent NO/cGMP/PKG pathway in the heart (Takimoto E, Champion HC, Belardi D, et al. *Circ Res.* 2005 Jan 7;96(1):100-9. doi: 10.1161/01.RES.0000152262.22968.72), and therefore reduction of cardiac afterload through Zn^{2+} -mediated cardiac positive inotropy and PKG-mediated anti-hypertrophic action are involved in cardioprotective effect of PPZ2.

These data and information are described in the Results (page 13, line 10) and Discussion (page 16, line 8).

In their previous Sci Rep paper, the authors speculate that the background of the mice may contribute to the differential outcome of the TRPC6-KO (Oda et al., 2017): “Using TRPC6-deficient 129/Sv background mice, we found that TRPC6 deletion failed to suppress pressure overload-induced LV dysfunction as well as oxidative modification of plasma Gpx3 protein, despite significant suppression of interstitial fibrosis.” If this is the case, comparative studies need to be done to clarify the potential impact of TRPC6 in heart failure. The background of the MLP-KO is not given.

[Response]

We investigated the effect of PPZ2 on three models of heart failure using C57BL/6J background mice (for MI model) and 129Sv background mice (for TAC model and MLP-KO model). PPZ2 showed a cardioprotective effect in TAC-operated WT mice, and this cardioprotective effect was not observed in TRPC6-deficient TAC-operated mice. These results are consistent with those of our previous report (Sci Rep, 2017), except the fact that TRPC6 deletion failed to attenuate TAC-induced cardiac fibrosis in this study. This difference may be partially because we here used slightly elder (28-

week-old) TRPC6-KO mice due to the limits of mouse breeding in time. However, it should be noted that PPZ2 dramatically improved post-TAC heart failure in a TRPC6 channel-dependent manner even under these conditions. From the results of our study, the cardioprotective effect by TRPC6 activation is considered to be a universal mechanism that does not depend on the mouse strain. These are described in the Methods (Supplementary Information page 2, line 16) and Results (page 13, line 10).

In Fig. 2, a close proximity of β 1AR and TRPC6 is proposed to explain the specificity of the proposed signaling pathway. Even though the pictures look nice, controls are missing for antibody specificities. β AR antibodies are rather problematic and especially the detection of endogenous β AR is rather difficult. Convincing antibody controls and possibly PLA assays of the overexpressed receptors with respective Taqs could contribute to the validation of the experiments, also Western blots for the base and apical region of the heart. Is the close β 1AR and TRPC6 proximity specific for β 1AR? Other interaction assays would help to substantiate this hypothesis.

[Response]

We assessed the specificity of PLA(TRPC6- β 1AR) using TRPC6^(-/-) mouse cardiomyocytes and found that PLA-positive signals were almost completely diminished in TRPC6^(-/-) cardiomyocytes (Supplementary Figure 2c). In β 1AR-GFP-overexpressing HEK293 cells, both GFP images and β 1AR immunostaining images were completely merged and showed positive signals on the plasma membrane (Supplementary Figure 2d). In addition, the signal intensities of β 1AR staining were dramatically reduced in isolated adult mouse cardiomyocytes transfected with siRNA for β 1AR (Supplementary Figure 2e). These results strongly support that the data obtained by the PLA assay with TRPC6 and β 1AR antibodies are reliable.

We examined whether GFP-tagged β 1AR physically interacts with Flag-tagged TRPC6 using immunoprecipitation assay. The β 1AR-GFP proteins co-precipitated with FLAG-tagged TRPC6 proteins in HEK293 cells (Supplementary Figure 2f). This indicates that β AR physically forms protein complex with TRPC6. Actually, overexpression of Snapin-PDZ-linker peptide enhances the PLA-positive interactions between β 1AR and TRPC6 in neonatal rat cardiomyocytes (Supplementary Figure 7e).

These data and information are described in the Results (page 7, line 24)

The authors show that ISO-induced inotropy and chronotropy are similarly affected in TRPC6-KO and TRPC3-KO (Fig.1c/d) as in response to baroreceptor-induced inotropy (Fig.1a/b). Same for Fig. 3, i.e. ISO-induced sarcomere contraction/Ca²⁺ cycling. This is somewhat puzzling since the authors – at least that is my impression – do not propose an

impact of basal TRPC6 physiological function. How can then the TRPC6-KO impact on ISO-stimulation if bAR are downstream of TRPC6 (also see scheme in Fig.7)? Can PPZ2 enhance the ISO-mediated effects? Is the TRPC6-phenotype unaltered if PPZ2 is applied? Are these effects in Fig1 and Fig3 really due to Zn²⁺ influx but not Ca²⁺ influx?

[Response]

The positive inotropic response induced by acute hypotension (baroreflex) or ISO treatment was weakened only in TRPC6-KO hearts, but not in TRPC3-KO hearts (new Fig. 1a–d).

Genetic deletion of *trpc6* causes a significant reduction of intracellular Zn²⁺ levels in several types of cells (Fig. 4a–c), and the overexpression of TRPC6 protein is sufficient to prevent the βAR-stimulated Gs/βarrestin-dependent βAR internalization in a Zn²⁺-dependent manner (Fig. 5 and Supplementary Fig. 6b). Zn²⁺ application enhances the norepinephrine (NE)-induced β1AR-Gs coupling (Supplementary Fig. 6a) and Zn²⁺ chelation attenuates cAMP production and cardiomyocyte contraction induced by NE with PPZ2 (Fig. 4k–m, Supplementary Fig. 7a, c). We identified a TRPC6 (KYD) mutant in which TRPC6-mediated Zn²⁺ influx activity, but not cation influx activity, is reduced (Fig. 4g and Supplementary Fig. 4e-m). The increase of cAMP production by ISO stimulation was significantly reduced in TRPC6^(KYD/KYD) cardiomyocytes compared with WT TRPC6 cardiomyocytes (Supplementary Fig. 5 and Fig. 4m). We also showed that long-time treatment with PPZ2 had no effect on TRPC6 localization (Supplementary Fig. 7d), excluding the possibility that the cardioprotective effect of PPZ2 is caused by TRPC6 downregulation. Our results can not completely rule out the potential involvement of Ca²⁺/Na⁺ influx in TRPC6-mediated enhancement of cardiac positive inotropy; however, only Zn²⁺ influx can explain the functional difference between TRPC3 and TRPC6. These data are described in the Results (page 6, line 9; page 9, line 12)

If the low levels of intracellular Zn²⁺ are thought to contribute to the contractile phenotype in response to ISO, it should be analysed whether the TRPC6-KO cells with long-term low Zn²⁺ level do not possibly suffer from ER stress or other dysfunctional proteins that may impact on ISO-stimulated Ca²⁺ cycling and sarcomere contraction? Increased intracellular Zn²⁺ levels/dysregulated Zn²⁺ homeostasis have been associated with increased ER stress and RyR2 leakiness in cardiomyocytes and are hypothesized to contribute to the development of heart failure (e.g. Reilly-O'Donnell et al., JBC, 2017 or Olgar et al., 2018). ER stress should be analyzed in the TRPC6-KO and the long-term mouse model.

[Response]

We examined whether TRPC6-KO cells from mice causes ER stress. We found that the mRNA expression levels of ER stress markers were similar in both young and old hearts of WT and TRPC6^(-/-) mice (Supplementary Fig. 3d, e). These data are described in the Results (page 9, line 21).

In Fig.4, it is shown that the beta-blocker propranolol antagonizes the α 1AR-TRPC6-Zn²⁺ axis. If this is the case, is TRPC6 a promising therapeutic target in heart failure since beta-blockers are given to most heart failure patients? This should be discussed at least. Further, PKG signaling is thought to be beneficial in heart failure, and PKG has also been shown to impact on TRPC6 and thereby to attenuate excess SSC and arrhythmia (Seo et al., Circ Res 2014).

[Response]

Thank you for meaningful comments. We added the following sentence in the Discussion: However, β -blockers are given to most heart failure patients, and we demonstrated that propranolol antagonizes the α 1AR-TRPC6-Zn²⁺ axis (Fig. 4h). This implies that the beneficial positive inotropic effect by TRPC6 activator is not expected in heart failure patients treated with β -blockers, but TRPC6-mediated Zn²⁺ influx may help reduce the load on the heart. A recent report showing that Zn²⁺ drives vasorelaxation by acting in sensory nerves, endothelium and smooth muscle also supports the physiological significance of intracellular Zn²⁺ as a new therapeutic target (Betrie et al., Nature Commun., 2021).

Interestingly, treatment with PPZ2 increased PKG activation in TAC-operated hearts (Fig. 6g). Previous studies reported that increased PKG activity prevents LV remodeling (i.e., hypertrophy and fibrosis) and dysfunction. β AR stimulation is reported to activate NOS3-dependent NO/cGMP/PKG pathway in the heart (Takimoto et al., Circ Res. 2005 Jan 7;96(1):100-9.). As the reviewer has pointed, we hypothesize that the PPZ2-induced PKG activation is a compensative negative feedback mechanism to attenuate excess stress-stimulated contractility and arrhythmia through TRPC6 in dystrophic hearts (Seo et al., Circ. Res., 2014). These are mentioned in the in the Results (page 13, line 10) and Discussion (page 16, line 8).

In Fig. 2, a close proximity of β 1AR and TRPC6 is proposed and it is also speculated that TRPC6 interacts with α 1AR in cardiomyocytes and that this contributes to the proposed mechanism (α 1AR-TRPC6-Zn²⁺- β 1AR). Is this a triple complex that is postulated? And why should TRPC6 not lead to a Ca²⁺ influx in cells that do not express β 1AR (Suppl Fig. 5b/c)? Does the complex contribute to the specificity of TRPC6 to certain ions? These hypotheses need to be clarified. The authors state that TRPC6 does not impact on bArr

recruitment to the AT1R, but can the AT1R as Gq coupled receptor also increase the responsiveness of β AR via TRPC6? Is it the proximity or snapin that leads to the proposed specificity of α 1AR for TRPC6?

[Response]

In addition to a close proximal interaction between β 1AR and TRPC6 using immunostaining, we could detect a physical interaction using immunoprecipitation assay (Supplementary Figure 2f). Previous reports suggest that α 1AR functionally couples with TRPC6, and here we demonstrated that TRPC6 could form a protein complex with β 1AR. Therefore, TRPC6 may form a signaling complex with α 1AR and β 1AR. To enhance α 1AR-TRPC6- β 1AR axis, we expressed Snapin-PDZ-linker peptide in cardiomyocytes. Overexpression of the Snapin-PDZ-linker peptide significantly potentiated the NE-induced contraction of cardiomyocytes (Supplementary Figure 7b). This result also supports our idea of functional coupling among α 1AR/TRPC6/ β 1AR proteins.

Regarding Supplementary Fig. 5b/c (Supplementary Fig. 4b/c in the revised version), we evaluated the potentiating effect of TRPC6-mediated Ca^{2+} influx activity on α 1AR-stimulated Ca^{2+} mobilization in HEK293 cells. This data also supports the functional coupling between α 1AR and TRPC6, and the α 1AR-stimulated TRPC6 activation mobilizes Ca^{2+} as well as Zn^{2+} in HEK293 cells. As TRPC6 is activated by diacylglycerol (DAG) produced through the $\text{G}\alpha\text{q}$ -PLC pathway, Gs-coupled β AR is not necessary for this activity. This indicates that the functional coupling among α 1AR/TRPC6/ β 1AR proteins does not contribute to the specificity of TRPC6-mediated Zn^{2+} influx, but the TRPC6-mediated mobilization of trace amount of Zn^{2+} (but not other cations) is essential for the enhancement of positive inotropic effects in cardiomyocytes. We have no way to disrupt the interaction among α 1AR/TRPC6/ β 1AR proteins; however, as β 1AR and TRPC6 could physically interact (Supplementary Figure 2f) and Snapin-PDZ-linker expression enhanced NE-induced cardiomyocyte contraction (Supplementary Figure 7b), we believe that triple complex formation is a requirement. Angiotensin II type I receptor (AT1R) is also a Gq-coupled GPCR and can activate TRPC6 through PLC-dependent DAG production in cardiomyocytes (Onohara N, Nishida M, Inoue R, et al. EMBO J. 2006 Nov 15;25(22):5305-16. doi: 10.1038/sj.emboj.7601417), and therefore Ang II stimulation will be able to activate the TRPC6- Zn^{2+} - β 1AR signaling pathway as well as α 1AR. We previously reported that TRPC6 channels are commonly activated by several Gq-coupled receptors, such as α 1AR, AT1R, ET-1 and P2Y2 receptors (Sunggip C et al. Front Pharmacol. 2018 May 22; 9:523. doi: 10.3389/fphar.2018.00523). We believe that complex formation is a requirement for the activation of the TRPC6- Zn^{2+} - β AR axis, but the activation of α AR

together with β AR in response to NE release from sympathetic nerve ending is essential for the efficient activation of β AR signaling in cardiomyocytes.

Further comments:

o Table 3:

- In table 3 the data of the echos after 1 and 2 weeks are missing. They need to be included.

[Response]

We have included data of the echos after 1 and 2 weeks in Supplementary Table 7 in the revised manuscript.

- The n-numbers are missing in the table. In case the n-number is “3”, the number is extremely low.

[Response]

In the revised manuscript, we have included data for $n \geq 5$ in the table.

- The data in table 3 rather suggest (difficult with $n=3$) an increase of cardiac hypertrophy by PPZ2 (Table 3; IVS 0.69 vs 0.81 and LVPW 0.72 vs 1.03), while the CSA (Fig,6f) shows a significantly reduced cardiomyocyte size. How can this be explained? Could a concomitant calcium influx be responsible for that?

[Response]

We have included data for $n \geq 5$ as indicated above. In the revised Supplementary Table 7, there was no significant difference in IVS and LVPW, and PPZ2 did not exacerbate myocardial hypertrophy.

- The type of mouse used needs to be mentioned in the legend.

[Response]

We have added information on the type of mice used in the legends.

- n-numbers should be given in all legends.

[Response]

We have included the N-numbers for all experiments in the legends.

- Quality of blots in Suppl Fig. 3 is low, especially for GRK2. This needs to be reduce and a control would be of help.

[Response]

Thank you for the helpful advice. Actually, we tried several GRK2 antibodies, but they didn't work. We reduced GRK2 immunoblot band from Suppl. Fig. 3, as the reviewer has suggested.

Responses to Reviewer 2.

1) There are several small molecules specific for TRPC6, which work in vivo. As an example, SAR-7334 (1), laryxil acetate (2), and BI 749327 (3) are active in the nanomolar range and have more than an order of magnitude of selectivity for TRPC6 over the closely related TRPC3 and TRPC7 channels and do not block more distantly related TRPC channels, including TRPC4 and TRPC5. Use any of these compounds in vivo should greatly benefit the current study and provide direct evidence of TRPC6 contribution.

[Response]

Thank you for your helpful comment. We used BI749327 as recommended and demonstrated that the PPZ2-induced enhancement of baroreflex-dependent cardiac positive inotropy was canceled by co-treatment with BI749327 (Supplementary Figure 7i). These data are described in the Results (page 13, line 6).

2) The authors clearly describe the TRPC6-mediated Zn²⁺ influx, but potential calcium influx through the same mechanism should be better addressed. The authors should test if the exact mechanism affects TRPC6-mediated Ca²⁺ influx in the same cells, which might also improve chronic heart failure in mice.

[Response]

Thank you for this helpful comment. Following the recommendation, we performed additional experiments and added new data that supports the requirement of TRPC6-mediated Zn²⁺ influx in the enhancement of cardiomyocyte positive inotropic response *in vitro*. We found that TRPC6-mediated inhibition of β Arr recruitment to β AR was attenuated by a Zn²⁺ chelator (TPEN) but not a Ca²⁺ chelator (EGTA) (Supplementary Fig. 6b). Treatment of isolated adult cardiomyocytes with TPEN decreased the ISO-stimulated cAMP production (Fig. 4l), and the increase of cAMP production by ISO stimulation was significantly reduced in TRPC6^(KYD/KYD) cardiomyocytes compared with WT cells (Supplementary Fig. 5 and Fig. 4m). While these results do not completely rule out the effect of Ca²⁺ influx, our data strongly suggest that α AR-mediated TRPC6 activation potentiates NE-induced cAMP production in cardiomyocytes primarily in a Zn²⁺-dependent manner. We have also included information on the possible involvement of TRPC6-mediated local Ca²⁺ signaling in the protection of the heart against oxidative stress in the Discussion (Nishida M, Onohara

N, Sato Y, et al. J Biol Chem. 2007 Aug 10;282(32):23117-28. doi: 10.1074/jbc.M611780200). These findings and discussion are described in the Results (page 12, line 6; page 11, line 6) and Discussion (page 15, line 11).

3) M085 and GSK1702934A, two small molecules, which directly activate TRPC6 through a mechanism of stimulation of extracellular sites formed by the pore helix and transmembrane helix (4). It would be great if the authors test one of these agonists on Zn²⁺ influx for both wild-type and mutated channels.

[Response]

Thank you for your helpful suggestion. We performed experiments using GSK1702934A as suggested and found that it significantly increased the amount of intracellular Zn²⁺ through TRPC6 WT channels but not the TRPC6 mutant (Supplementary Fig. 4l, m). These results are described in the Results (page 11, line 2).

4) Some experiments have only n=3 (f.i., data reported in Figure 6), which appears a small number of experiments.

[Response]

In the revised manuscript, we have included data for n \geq 5 for all *in vivo* experiments.

5) Multiple cells lines and rodents (both mice and rats) were utilized for these experiments. The authors should provide some brief rationale for the selection of specific cell/animal models (technical limitations, etc).

[Response]

Considering statistical power for multiple comparisons of *in vivo* experiments, we used mice that can breed in large numbers at the same time and easy to genetically modify. To exclude the possibility of artifacts from strain differences, the pharmacological effect of PPZ2 was also evaluated using mice of C57BL6J mice (for MI) in addition to 129Sv mice. Because acutely isolated mouse cardiomyocytes are difficult to maintain in long-term culture, experiments in cultured cells were mainly performed using high-isolation-yield rat cardiomyocytes. HEK293 and RAOSMC cells are easy-to-use cell lines for transient expression of multiple proteins simultaneously and stable expression of TRPC6, respectively. The TRPC6-mediated Zn²⁺ influx activity was measured using subculture-able cells isolated from TRPC6^(-/-) mice, such as splenocytes and vascular smooth muscle cells, to confirm the universality of the Zn²⁺ pool controlled by TRPC6. This information has been included in the Methods (Supplementary Information page 18, line 2).

6) Data shown in supplementary figures 4a-4e could be added to the main figures if space allows.

[Response]

We have added these data as new Figure 4 in the revised manuscript.

7) Recent study about zinc as a target for vascular therapeutics could be discussed (Betrie et al., Nat Commun, 2021; PMID: 34075043).

[Response]

We appreciate the helpful suggestion; we have included discussion on the therapeutic potential of Zn²⁺ for vascular diseases in the revised manuscript. This text is included in the Discussion (page 15, line 4).

Responses to Reviewer 3.

1. Figure 1a,b nicely shows that the deletion of TRPC6 reduces the positive inotropic (but not chronotropic) effect of Hydralazine. How about TRPC3 knockout under these conditions?

[Response]

Thank you for your comment. We performed experiments using TRPC3 knockout mice and found that hydralazine-induced positive inotropy and chronotropy were similar between WT and TRPC3^(-/-) hearts (Fig. 1a, b). This information has been included in the Results (page 6, line 7).

2. Figure 2 shows nicely that TRPC6 co-localizes with beta1-AR in LV but not SAN cells. Can the authors comment in the discussion on the possible localisation of TPRC6 to submembrane structures of myocytes? As far as I remember there was some literature that TRPCs can be found in T-tubules of myocytes. If the co-localisation of beta1-AR takes place there, these could explain why SAN cells (which have less T-tubules per se) do not show this phenomenon. If the authors want to go in more detail here experimentally, one could check if chemical detubulation of myocytes or cholesterol depletion would reduced PLA signal.

[Response]

Thank you for this insightful comment. PLA puncta were abundantly detected in T-tubules of isolated cardiomyocytes. We used formamide as a detubulation agent and observed reduced PLA signals in isolated cardiomyocytes (Fig. 2c). These data are described in the Results (page 7, line 24). According to the reviewer's suggestion, we commented the difference results in SAN and LV cells because of reduced T-tubules in SAN cells in the Results (page 8, line 3).

3. Having said that it is unfortunate that cAMP FRET measurements were performed in neonatal instead of adult myocytes since neonatal cells do not have a mature well organised membrane structure including T-tubules. I will highly appreciate if such measurements could be done in adult cells, though I understand that an adenovirus would be required for transduction of these cells with the sensor. If this is not technically possible for the lab, maybe they could do cAMP ELISA on adult WT vs KO myocytes. It is also not well discussed in the text how the authors explain the reduced cAMP response to ISO in TRPC6 KO cell in Fig 3e,f. ISO is a beta-selective agonist which does not activate alpha-receptors. How does the alpha-AR-TRPC6-Zn connection feeds into here or it is just the "basal" TRPC6 activity which matters in this case?

[Response]

Because of technical limitations to perform adenovirus experiments in adult cardiomyocytes, we performed cAMP ELISA assays as suggested. Treatment of isolated adult mouse cardiomyocytes with PPZ2 increased the ISO-stimulated cAMP production (Fig. 4k). Treatment of cardiomyocytes with TPEN decreased the ISO-stimulated cAMP production (Fig. 4l). The increase of cAMP production by ISO stimulation was significantly reduced in TRPC6^(KYD/KYD) mutant cardiomyocytes compared with WT cardiomyocytes (Supplementary Fig. 5 and Fig. 4m). We also show that the pooled Zn²⁺ amount in TRPC6^(-/-) cardiomyocytes was smaller than amounts in WT and TRPC3^(-/-) cardiomyocytes (Fig. 4a-c). As the increase in TRPC6 protein expression was also sufficient to suppress ISO-stimulated β AR internalization in HEK293 cells and genetic TRPC6 deletion reduces intracellular Zn²⁺ amount in several cell types, basal TRPC6 activity is important to control intracellular Zn²⁺ amount, especially in TRPC6-overexpressing HEK293 cells. However, our findings strongly suggest that receptor-stimulated endogenous TRPC6 channel activation positively regulates β AR-stimulated cardiomyocyte contraction in a Zn²⁺-dependent manner. These data are described in the Results (page 11, line 5).

Calcium measurements in Fig 3 show data analysis only on the peak height which is inotropic response, did the authors analyse also the rate of Ca or relaxation decay, in other words does TRPC6 affect the positive lusitropic effect of ISO and/or cell relaxation at basal state?

[Response]

Thank you for your comment. As suggested, we analyzed the lusitropic effect of ISO in isolated mouse cardiomyocytes and mouse hearts. ISO-induced positive inotropy, chronotropy, and lusitropy were similarly induced in WT and TRPC3^(-/-) mice, while

ISO-induced positive inotropy and lusitropy, but not chronotropy, were impaired in TRPC6^(-/-) mice (Fig. 1c–e). The increase in sarcomere length shortening amplitude and the relaxation decay by ISO stimulation were significantly reduced in TRPC6^(-/-) cardiomyocytes compared with WT cells (Fig. 3a, b). There were no significant changes in cell relaxation rate at the basal state. These findings are described in the Results (page 6, line 17; page 8, line 22).

4. The choice of MLP mouse models as a dilated hypertrophy model is not well justified. The authors should provide a clear rationale why this mouse model was chosen for the study. Will the protective effect of the peptide be present also in other classical heart failure model such as after pressure overload or myocardial infarction?

[Response]

We appreciate the helpful comment. To address this issue, we performed extensive experiments using three different mouse models for heart failure: the transverse aortic constriction (TAC), myocardial infarction (MI), and MLP-deficient dilated cardiomyopathic model mice. Treatment of TAC-operated mice with PPZ2 significantly improved cardiac contractility, cardiomyocyte hypertrophy, and fibrosis in a TRPC6 inhibition-dependent manner (Fig. 6a–d and Supplementary Table 3,4). In addition, PPZ2 treatment significantly increased intracellular Zn²⁺ and PKG activity in TAC-operated hearts (Fig. 6e–g). Treatment of MI and MLP-KO mice with PPZ2 significantly improved cardiac contractility, cardiomyocyte hypertrophy, and fibrosis (Fig. 6h, i and Supplementary Fig. 8a–f and Supplementary Table 5–8). These data are described in the Results (page 13, line 10) and Discussion (page 16, line 8).

5. Reference 26 appears two times to quote different studies. The first one referring to Takotsumo cardiomyopathy might be Paur H et al. Circulation 2012?

[Response]

Thank you for the careful review of our manuscript. We have corrected this error.

6. Mechanistically the connection between Zn and beta1-AR/Gs/cAMP pathway is referred to largely based on previously published literature showing that Zn can effect Gs, GRK activities and receptor affinity for ligands such as ISO or NE, no experiments on it were done for this particular paper. I feel.

[Response]

We added in vitro experiments and confirmed that the increase in BRET intensity between β 1AR and Gs by NE stimulation was enhanced by external Zn²⁺ application (Supplementary Fig. 6a). These are described in the Results (page 11, line 10)

7. Figure 6a,b - there is a discrepancy between positive inotropic response to NE in control vs RFP expressing cells (3.5 vs 2 fold increase), and almost no effect of NE in Fig 6c control column. Can the authors explain this discrepancy?

[Response]

This difference is because of the different NE concentrations used for inducing positive inotropic response of cardiomyocytes in the experiments. In Supplementary Figure 7a–b (previous Fig. 6a–b), cardiomyocytes were treated with NE at a concentration of 1 μ M. The reduction of NE-induced contraction in cardiomyocytes in 7b compared with that in 7a may be because of transfection stress, which may suppress the response to NE. Overexpression of Snapin-PDZ-linker completely recovered the NE-induced contraction. In Supplementary Figure 7c (previous Fig. 6c), cells were treated with lower concentrations of NE (at 0.1 μ M). This information is listed in the legend.

8. Fig. 4d - cells were pretreated with receptor blockers for 24 or 25 h. Why that long? The blockers should work normally almost instantaneously.

[Response]

This experiment was carried out simultaneously with the α 1AR activator treatment (Fig.4i) as front and back experiments. Since the treatment time of the α 1AR activator was first examined and the preliminary data showed that the activator had a greater response to the 24 h treatment than the 1 h treatment, we set the treatment time to 24 h. To align the front and back experimental conditions, the inhibitor treatment time was also set to 24 h. As it was obvious that β AR blockers rapidly suppressed cAMP production, we did not conduct comparative experiments between short-term and long-term treatments.

REVIEWERS' COMMENTS

Reviewer #1 (Remarks to the Author):

The authors have addressed all my concerns. Congratulations to the profound and elegant study with great impact in the understanding of the molecular mechanisms in heart failure therapy.

Reviewer #2 (Remarks to the Author):

The authors mostly addressed all previously raised concerns. My only suggestion is to include Supplementary Figure 7i as part of the main figures.

Reviewer #3 (Remarks to the Author):

The authors have addressed all my concerns satisfactorily. Their manuscript is acceptable for publication in its present form

Responses to Reviewers.

Reviewer #2 (Remarks to the Author):

The authors mostly addressed all previously raised concerns. My only suggestion is to include Supplementary Figure 7i as part of the main figures.

[Response]

Thank you for your helpful comment. We moved the result of previous Supplementary Figure 7i to main Figure 6a in the present version.